



# Evaluation of the New NDACC Ozone and Temperature Lidar at Hohenpeißenberg and Comparison of Results with Previous NDACC Campaigns

Robin Wing[1], Sophie Godin-Beekmann[1], Wolfgang Steinbrecht[2], Thomas J. McGee[3], John T. Sullivan[3], Sergey Khaykin[1], Grant Sumnicht[3], and Larry Twigg[3]

[1]LATMOS/IPSL, OVSQ, Sorbonne Universités, CNRS, Paris, France
[2]Deutscher Wetterdienst, Met. Obs. Hohenpeißenberg, Hohenpeißenberg, Germany
[3]NASA Goddard Space Flight Center, Greenbelt, Maryland

**Correspondence:** Robin Wing (robin.wing@latmos.ipsl.fr)

**Abstract.** A newly upgraded German Weather Service (DWD) ozone and temperature lidar (HOH) located at the Hohenpeißenberg Meteorological Observatory (47.8° N, 11.0° E) has been evaluated through comparison with the travelling standard lidar operated by NASA's Goddard Space Flight Center (NASA STROZ), satellite overpasses from the Microwave Limb Sounder (MLS), the Sounding of the Atmosphere using Broadband Emission Radiometry (SABER), the Ozone Mapping and Profiler Suite (OMPS), meteorological radiosondes launched from München (65 km north-east), and locally launched ozonesondes.

The "blind" evaluation was conducted under the framework of the Network for the Detection of Atmospheric Composition Change (NDACC) using 10 clear nights of measurements in 2018 and 2019. This campaign was conducted within the larger context of NDACC validation activities for European lidar stations. The previous 2017-2018 validation campaign took place at the French Observatoire de Haute Provence and and showed a high degree of fidelity between participating instruments. The results are reported in the companion article (Wing et al., 2020).

There was good agreement between all ozone lidar measurements in the range of 15 to 41 km with relative differences between co-located ozone profiles of less than ±10%. Differences in the measured ozone numbers densities between the lidars and the locally launched ozone sondes were also generally less than 5% below 30 km. The satellite ozone profiles demonstrated some differences with respect to the ground based lidars which are due to sampling differences and geophysical variation.

Temperatures differences for all instruments were less than ±5K below 60 km, with larger differences present in the lidar-satellite comparisons above this region. Temperature differences between the lidars met the NDACC accuracy requirements of ±1 K between 17 and 78 km. The NASA lidar exhibited slightly colder temperatures, between 5 and 10 K, than the other instruments below 20 km and slightly warmer temperatures, 5 to 10 K, above 70 km. These differences are likely due to algorithm initialisation choices and photon count saturation corrections.





## 1 Introduction

The Network for the Detection of Atmospheric Composition Change (NDACC, http://www.ndacc.org) is an international collaboration of more than 70 research stations (Kurylo et al., 2016; De Mazière et al., 2018) which provides a common framework for the early detection of long-term changes in the atmosphere and validation of atmospheric measurements. To facilitate these instrument validation exercises, a mobile reference lidar operated by NASA's Goddard Space Flight Center

(NASA STROZ) is shipped around the world to conduct intensive comparison campaigns with other NDACC lidars. Most recently, NASA STROZ participated in the LAVANDE campaign at the Observatoire de Haute Provence in southern France LAVANDE.

A general background on ozone lidars, analysis techniques, data collection procedures, and NDACC comparison parameters of this study mirror the work recently done during LAVANDE. For the purposes of this article we will endeavour to provide brief

but comprehensive introductions on each of these topics without engaging in onerous repetition. We invite readers seeking more details of NDACC lidar validation activities to consult the companion paper Wing et al. (LAVANDE, 2020) and other NDACC studies: (STOIC, Margitan et al., 1995); (OPAL, McDermid et al., 1998); (OTIC, Braathen et al., 2004); (NAOMI, Steinbrecht et al., 1999); (HOPE, Steinbrecht et al., 2009); (MOHAVE, Leblanc et al., 2011); NDACC algorithm intercomparisons for ozone lidars (Godin et al., 1999); as well as a review paper summarising NDACC validation exercises (Keckhut et al., 2004).

When providing context for this most recent validation exercise we will refer back to the 2020 LAVANDE study and the last validation study at Hohenpeißenberg, the during the HOPE campaign published in 2009.

In general, NDACC lidars measure stratospheric ozone with an accuracy better than 3% between 12 and 35 km altitude and better than 10% between 35 and 40 km. NDACC lidar temperature measurements similarly have an accuracy better than 1 K from 30 to 40 km altitude when compared with co-located measurements. Lidar precision for ozone is highest near the

peak concentration of ozone in the stratosphere and decreases above and below the layer as the signal to noise ratio drops at low ozone concentrations. The precision for temperature typically decreases above 70 km depending on the the laser power, telescope area, and integration time for a given lidar measurement. Further details on the theoretical uncertainty budgets for NDACC temperature and ozone lidars can be found in (Leblanc et al., 2016a, b, c).

### 1.1 Key Results from HOPE

The previous NDACC validation campaign (HOPE, Steinbrecht et al., 2009) found a low bias in the ozone profiles produced by the Hohenpeißenberg Original (HOHO) lidar between 33 and 43 km by up to 10% and a high bias of approximately 50% above 50 km when compared with the travelling standard lidar operated by NASA-STROZ. These differences were attributed to the choice of numerical filters used by the NASA and DWD algorithms. An investigation of the precision for ozone data from both lidars concluded that the agreement between profiles from each system was better than 5% between 20 and 40 km.

The 2009 HOPE campaign study also found that the HOHO lidar measures temperatures 1 to 2 K colder than the NASA lidar between 30 and 65 km and up to 15 K warmer than NASA above 65 km. These differences were only significant from 25 to 50km. Additionally, a small altitude offset of 290 m was discovered and corrected in the HOHO system.





## 1.2 Key Results from LAVANDE

During the recent NDACC validation campaign by Wing et al. (LAVANDE, 2020) there was good agreement between all
ozone measurements between 20 and 40 km with differences of less than 5% throughout this region. There were no statistically
significant differences (at the 95% confidence level) between the NASA-STROZ reference lidar and the French OHP LiO$_3$S
lidar between 18 and 39 km. Above and below this region the percent differences increase. It is important to note that the
differences in the absolute number density of the measured ozone remains low as these regions are well removed from the peak
of the ozone layer. Agreement between the lidars and the satellites MLS and SABER were also good to within 10% between
20 and 40 km. SABER tended to produce unrealistic ozone values below 20 km. The ECCs were in statistical agreement (at
the 95% confidence level) with both lidars between 15 and 30 km. Above 30 km the uncertainties associated with the pump
correction at low pressures contributed to larger measurement differences.

The temperature measurements of the NASA-STROZ reference lidar and the OHP lidar LiO$_3$S were statistically equal from
22 to 60 km. Temperature is a secondary scientific product for LiO$_3$S which is currently not archived with NDACC or reported
above 60 km. A comparison was also conducted between NASA and the OHP temperature lidar LTA. The validation exer-
cise determined that the photomultiplier in the low gain channel of LTA was defective and the component was subsequently
replaced. NASA exhibited an apparent cold bias of approximately 3 K below 25 km with respect all other instruments. Temper-
ature agreement between the lidars and the satellites MLS and SABER were generally very good throughout the stratosphere,
only exceeding ±5 K above 55 km. MLS exhibited a vertical oscillation in the temperature profiles with an amplitude of ±5 K
with respect to all other measurements. The characteristics of this MLS-lidar difference have been previously reported in Wing
et al. (2018b). The ECC and radiosondes were also in agreement with the lidars.

Total uncertainty estimates for ozone and temperature were calculated for each instrument involved in the campaign. This
was done in an effort to characterise the uncertainty budgets of each of the participating instruments with respect to the observed
standard deviation between each set of measurements. This comparison allowed us to evaluate the uncertainty estimates for the
lidars and determine if we are realistically estimating the measurement uncertainty in our instruments and the total uncertainty
in our profiles of ozone and temperature. We found two outstanding issues during this exercise: 1) the temperature uncertainty
budget for the LiO$_3$S lidar overestimates the uncertainty above 35 km 2) there was a previously undetected discrepancy between
the temperature uncertainty budget for the French lidar LTA and NASA of up to 2 K below 50 km. In response to the LAVANDE
campaign findings the PMTs for the low gain channels (< 50 km) in LTA were replaced and plans were made to modify filtering
codes for LiO$_3$S temperatures for eventual submission to the NDACC database.

## 1.3 Article Overview

The Hohenpeißenberg Ozone Profiling Study (HOPS) campaign took place in October 2018 and March/April 2019 (see Tab.
2) with the dual purpose of providing an updated validation of the existing DWD ozone lidar, hereafter referred to as Hohen-
peißenberg Original (HOHO), which has been in continuous operation since September 1987 (see key instrument publications:
(Geh, 1987; Claude et al., 1994; Steinbrecht et al., 1997, 2009)) and a first validation study for the new and improved DWD



ozone lidar, hereafter referred to as Hohenpeißenberg lidar (HOH). A technical comparison of both instruments is given in Sect. 2.2. The work presented in this article follows the NDACC standards for 'blind' instrument intercomparisons. The measurements were made onsite and ozone and temperature profiles were calculated by the respective NASA and DWD lidar teams, the nightly averaged lidar profiles were collected by an impartial NDACC referee (S. Godin-Beekmann) who was not involved

in conducting the measurement campaign, and the intercomparison of the results was conducted by the referee's team.

The paper is structured according to the following outline: Sect. 2 introduces the instruments involved in the HOPS campaign and sets the co-location criteria for coincident measurements; Sect. 3 provides technical details for the new DWD temperature and ozone lidar and shows some examples of co-located ozone and temperature profiles; Sect. 4 conducts a statistical intercomparison between all instruments for ozone; Sect. 5 conducts a statistical intercomparison between all instruments for

temperature; Sect. 6 examines and assesses the estimated uncertainty budgets for all instruments participating in the HOPS campaign; Sect. 7 conducts a cross-intercomparison of both the LAVANDE and HOPS NDACC campaigns to assess the performance of the travelling standard lidar NASA-STROZ; and Sect. 8 summarises the major finding of the HOPS NDACC intercomparison campaign as well as the results of the LAVANDE-HOPS cross-comparison and evaluation of NDACC lidar validation activities in Europe.

## 2   Instruments used for HOPS

Table 3 summarises all the different systems participating in the HOPS intercomparison. Key aspects of each different instrument are noted in each subsection. References to original or most recent instrument descriptions are given for those seeking further details and can also be found in Wing et al. (LAVANDE, 2020).

### 2.1   Original DWD Lidar (HOHO)

The original DWD ozone lidar (HOHO) located at the Hohenpeißenberg Meteorological Observatory (47.8 N, 11.0 E) has been in continuous operation since 1987 and has one of the longest and most complete data records in NDACC. The lidar uses a differential absorption (DIAL) technique which exploits the scattering cross-sections for ozone at two different wavelengths. The first wavelength is generated using a Xenon Chloride excimer laser to generate a primary emission at 308 nm. The light passes through a hydrogen ($H_2$) gas cell where the primary emission is used to stimulate a Raman emission at 353 nm. Both

wavelengths are transmitted through a 10X beam expander to reduce the divergence of the laser beam before transmission to the sky. The receiver telescope is a 0.6 m Newtonian mirror. The 353 nm line is weakly absorbed by ozone (also referred to as the non-absorbed line or off-line) and can be used to infer the neutral density of the atmosphere above the aerosol layers present in the lower stratosphere. The shorter 308 nm line is more strongly absorbed by ozone (also referred to as the absorbed line or the on-line) and is used to detect the number of ozone scattering targets in a profile above the lidar. The DIAL technique uses

these two profiles to infer the ozone number density by taking the derivative of the ratio between these two measured profiles (Pelon and Megie, 1982). Generating lidar temperature profiles is accomplished using the Rayleigh lidar returns from the 353 nm channel. Relative density profiles can be inferred from the range corrected lidar proton counts profile. Using an assumed



a priori pressure at the top of the lidar profile, an absolute temperature profile can be calculated based on the relative density gradient. Full details for this technique are found in Hauchecorne and Chanin (1980). Below approximately 27 km both DWD lidars incorporate information from the local meteorological radiosonde in an effort to identify and correct for the possible contamination by stratospheric aerosol layers.

Full technical specifications can be found in Steinbrecht et al. (2009) and a comparison of the technical specifications of the original and new DWD ozone lidar can be found in Tab. 4.

The data processing for the HOHO lidar is as described in Steinbrecht et al. (2009). Lidar return signals are corrected for photon counter dead-time effects, the background is subtracted, and the signals are averaged over the night. After correction, the high gain and attenuated low gain signals are merged. Typically, the high gain signal is useful down to about 20 km and the low gain signal continues down to about 10 km. From the combined signals, temperature and ozone profiles are derived. Ozone profiles typically extend down to 10 or 15 km, depending on the night, while pure lidar temperature profiles care calculated down to 28 km (where aerosol becomes important and biases the retrieved temperature). The HOHO ozone algorithm uses a very wide differential filter in the ozone calculation (Godin et al., 1999; Steinbrecht et al., 2009). There is a resulting bias from the differential filter, which is substantial near 35 km and is corrected (see Steinbrecht et al. (2009)). Corrections for signal-induced noise and timing delay are also applied in the ozone processing code as required.

## 2.2 New DWD Lidar (HOH)

The newly upgraded DWD lidar also exploits the DIAL technique for measuring ozone. The key difference in the new system is the use of two lasers to generate the weakly and strongly absorbed lines in place of a Raman gas cell. The weakly absorbed line is generated at 355 nm from the frequency tripled output of an Nd:YAG laser and the second wavelength at 308 nm is produced using an excimer gas laser. In addition to using two dedicated high powered lasers to produce the lidar emissions, the new HOH lidar employs a 1 m receiver telescope, dedicated high and low gain channels at both 355 nm and 308 nm to improve the dynamic range of the lidar measurements, Raman channels at 332 nm and 387 nm, as well as new fast response PMTs. A full list and comparison for the technical specifications of the HOH system can be found in Tab. 4. A secondary objective for this paper is to characterise the measurement bias and uncertainty budget of the new HOH with respect to the HOHO to ensure continuity and consistency in the Hohenpeißenberg NDACC data record.

Data processing for the new HOH lidar is essentially the same as for the HOHO lidar. The vertical resolutions of the derived ozone and temperature profiles (and the differential filter for ozone) are the same for HOH and HOHO. The different instrumental parameters (faster counters, better timing, etc.) are accounted for in the processing. Due to the much better return signals, merging between low and high gain returns occurs at higher altitude, around 25 to 30 km. Precision of the measured ozone and temperature profiles is also better than for the HOHO lidar. Ozone profiles from the new HOH lidar usually cover the altitude range from 15 to 50 km (10 to 45 km for the old HOHO). Temperature profiles cover 28 to 80 km (28 to 65 km for the old lidar).



### 2.2.1  NASA Stratospheric Ozone Lidar (NASA STROZ)

NASA's Goddard Space Flight Center Stratospheric Ozone Lidar (NASA STROZ) the mobile NDACC validation lidar for temperature and ozone measurements. This mobile lidar system is shipped across the world and used to run intercomparison and validation campaigns for lidar stations within the NDACC network. The NASA STROZ is a DIAL system similar to the HOH, relying on an on-line wavelength of 308 nm and an off-line wavelength of 355 nm generated by two separate lasers. The system also has two Raman channels at 332 nm and 407 nm for tropospheric measurements. The system was constructed in 1988 (McGee et al., 1991) and has participated in many NDACC lidar campaigns for lidar stations around the world (McGee et al., 1995).

### 2.2.2  Radiosondes and Brewer-Mast ozonesondes (BM)

Brewer-Mast ozonesondes (BM) manufactured by Mast Keystone Co. consist of a single electrochemical cell with a silver anode and platinum cathode which are immersed in a potassium iodide (KI) solution (Solar and SIMS, 2014). The ozonesondes are attached to a to Vaisala RS92-SGP radiosondes and were launched approximately every two nights during the campaign. A total of five in-situ ozone measurements were made to compare with ten nightly average lidar profiles. Brewer-Mast ozonesonde uncertainty estimates are given as $\pm(3–5)\%$ by Stübi et al. (2008) however, we found this estimate to be too conservative and have adapted the uncertainty estimates for ECCs of $\pm(2.5–10)\%$ given by Tarasick et al. (2016).

In addition to the BMs, we have also used the Vaisala RS41-SGP meteorological radiosondes launched from the nearby station at München.

### 2.2.3  Microwave Limb Sounder (MLS)

The Microwave Limb Sounder (MLS) uses a spectrometer to make limb measurements of thermal microwave radiation of the atmosphere. The instrument, aboard the Aura satellite, allows for the retrieval of stratospheric ozone profiles with a vertical resolution of about 3 km. Measurements of stratospheric temperature profiles are alos made with a typical vertical resolution of 8 km at 30 km altitude, 9 km at 45 km altitude, and 14 km at 80 km (full width at half maximum (FWHM) of the averaging kernels,  Schwartz et al., 2008). MLS profiles of temperature, geopotential height and ozone were extracted from the Version 4.0 MLS dataset. A more complete description of the instrument is given in Waters et al. (2006). For the HOPS campaign, the geopotential altitude is converted to a geometric altitude and re-gridded to allow for a direct comparison with the lidars and sondes.

### 2.2.4  Sounding of the Atmosphere using Broadband Emission Radiometry (SABER)

Ozone and temperature measurements from the Sounding of the Atmosphere using Broadband Emission Radiometry (SABER) instrument were downloaded from 15 to 100 km. The vertical resolution for SABER temperature profiles is approximately 2 km and the estimated accuracy is 1 to 2 K between 15 and 60 km which decreases to 5 K near 85 km, and to 10 K near 100 km (Rezac et al., 2015a, b). Precision estimates for SABER ozone profiles are reported as 1% between 40 and 50 km



altitude, decreasing to 2% between 30 and 55 km and 10% near 80 km (Rong et al., 2009). A more complete description of the instrument is given in Mertens et al. (2001). SABER profiles of temperature, geopotential height and ozone were extracted from the Version 2.0 SABER dataset.

### 2.2.5 The Ozone Mapping and Profiler Suite (OMPS)

The Ozone Mapping and Profiler Suite Limb Profiler (OMPS-LP) on the Suomi National Polar-orbiting Partnership (Suomi-NPP) satellite, which has been in operation since April 2012, measures solar radiances scattered from the atmospheric limb in the ultraviolet (UV) and visible (VIS) spectral ranges. The VIS measurements are used to retrieve ozone in the lower stratosphere and are made using an ozone sensitive measurement at 602 nm coupled with two weakly absorbing lines at 510 nm and 673 nm. The UV measurements cover the middle and upper stratosphere up to 60 km altitude and exploit three
principle wavelengths at 302 nm, 312 nm, and 322 nm with the reference line at 353 nm. The LP sensor has three slits separated horizontally by 4.25° (about 250 km), which serves to expand the cross-track coverage. The vertical sampling of OMPS-LP measurements is $\sim 1$ km, although the actual instrumental field of view is about 1.3–1.7 km (Flynn et al., 2006, 2014). The given vertical resolution for ozone profiles in the stratosphere is 1 km.

The estimated uncertainty on the visible OMPS-LP ozone profile is given as a function of altitude and ranges from approxi-
195 mately 40% near 10 km, to 15 % at 20 km, to roughly 3 to 5 % in the rest of the stratosphere. The estimated uncertainty of the UV channel is approximately 4 % at 25 km and drops to 2.5 % at 35 km and is less than 2 % up to 60 km (Loughman et al., 2005; Zawada et al., 2018). In this study we use version 2.5 OMPS-LP ozone profiles described in Kramarova et al. (2018).

OMPS ozone profiles were not included in the LAVANDE study as at the time the authors considered the temporal offset too large. In HOPS we are making a first attempt at using a solar limb scanning satellite to validate night time lidar measurements.

### 200 2.2.6 Co-locating satellite profiles and ground-based profiles.

For HOPS, we considered all satellite profiles with a tangent point within ±5° latitude and ±15° longitude of the Hohenpeißenberg Meteorological Observatory (47.8° N, 11.0° E), and within ±6 hours of 00 UTC (1 hour after local midnight for the lidar measurements nights) for SABER, ±99 minutes of 1h40 UTC for MLS, and ±101 minutes of 11h50 UTC the following day for OMPS. This fairly large coincidence box is depicted in Fig. 1. It covers most of central Europe, from Wales in the northwest
to Bulgaria in the southeast. The box size chosen here is similar to the compromise chosen in Wing et al. (2018b) and relates to the trade off between a small number of close overpasses and a larger number of overpasses which may be further away from the ground station. For HOPS there are typically between 10 to 20 coincident profiles for each of the satellites, which are generally divided between one or two satellite overpasses, for a given night (the following morning for OMPS).





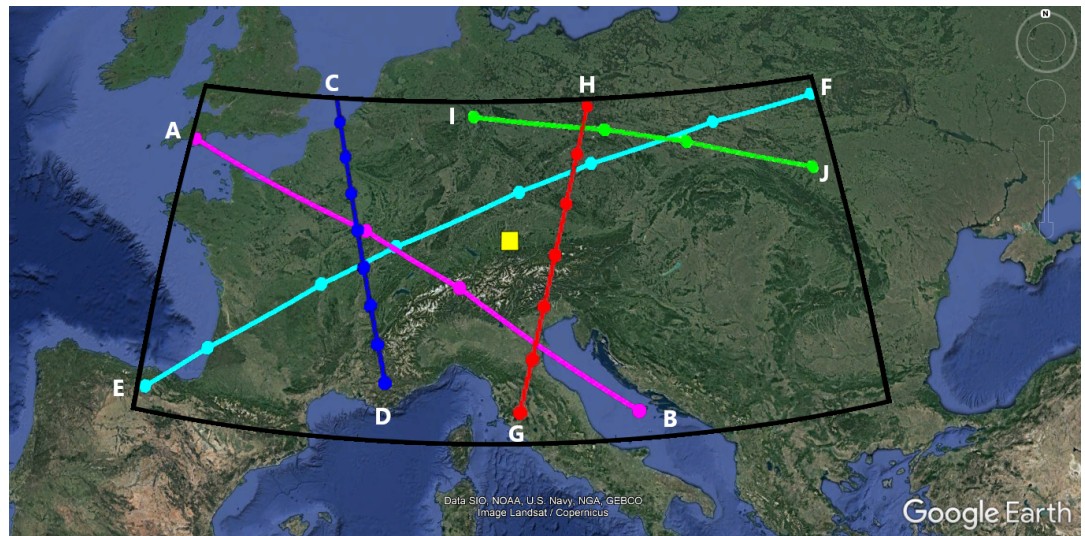

**Figure 1.** The area defined for coincident measurements during the HOPS campaign (42.8,-4.0) to (52.8,26.0). Meteorological Observatory Hohenpeißenberg is represented by the yellow square at (47.8,11.0) and the nearby München-Oberschleissheim radiosonde launches are located at (48.1,11.3). On the night beginning on 29 March 2019 the HOH lidar operated from 21:28-4:40 UTC; the MLS overpass (G-H, red) occurred at approximately 1:14 UTC on March 30th; the SABER overpasses occurred on the 29th at 22:45 UTC (E-F, cyan) and 23:30 UTC (A-B, magenta), and on March 30th at 1:16 UTC (I-J, green); and the OMPS overpass (C-D, blue) occurred on March 30th at 11:50 UTC. (data: © Google Earth Pro, 2020)

## 3 DWD NDACC Lidar Upgrades and Example Data

The HOPS campaign took place in two parts: the first period covered the nights of October $21^{st}$ and $22^{nd}$ 2018, and the second period covered nights in 2019, from March $21^{st}$ to April $6^{th}$. Table 2 shows which systems provided ozone and/or temperature profiles on each of the different nights of the campaign. Table 3 shows the details of the altitude range and important wavelengths for each instrument when making measurements of temperature and ozone.

### 3.1 Evaluation of the New HOH Lidar

The HOPS campaign provided a perfect opportunity to conduct an evaluation of the newly installed HOH lidar. The HOH lidar ran concurrently with the NASA-STROZ mobile validation lidar as well as the original HOHO lidar. This crucial overlap period allows us the opportunity to conduct a formal NDACC evaluation of the Hohenpeißenberg lidars and ensure that there are no unexplained biases or problems which could go on to cause discontinuities in one of the longest running NDACC datasets (1987-2020). Table 4 shows an in-depth comparison of the technical specification for the HOH, HOHO, and NASA lidars.

Figure 2 shows a comparison of the nightly average deadtime corrected, photon count rates for the high gain channels of the HOH and HOHO lidars as well as the ratio of the high and low gain channels for both systems. The left hand panel shows the photon count rates (PCR) for the high gain channels in both lidars at 308 nm and 355 nm. The signal in the 308 nm





channel for the HOH lidar (red) is 74x larger than the signal in the 308 nm channel of HOHO (blue). Similarly, the high gain 355 nm channel of HOH (green) has 224x more signal than the 353 nm high gain channel in HOHO (magenta). The signal improvements in the low gain channels at both wavelengths are not indicative of the general increase in system performance as there are neutral density filters placed in front of the photomultipliers to attenuate the signals. The increased SNR at of the HOH system with respect to the HOHO system results in ozone profiles with less statistical uncertainty (discussed later in Sect. 6.1) and the large 224 factor improvement in the high gain 355 nm channel will allow for Rayleigh temperature profiles to routinely reach the Upper Mesosphere and Lower Thermosphere (UMLT).

The right hand panel of Fig. 2 shows that there is significant improvement in the SNR of the high gain 308 nm channel (dark red) above 55 km. The high gain channel at 355 nm (dark blue) is linear over the entire altitude range. The ratio between the low gain channels for both 308 nm (dark green) and 355 nm (dark purple) have small slopes which indicates very slight offsets in the slopes of the PCR profiles. It is recommended that either the attenuation of the low gain channels be reduced or that the high gain channels be truncated at a lower altitude to provide a greater overlap region between the high and low gain channels where both have high SNR.

In the crucial range between 30 to 50 km for ozone and 30 to 70 km for temperature the HOH channels are linear with respect to their HOHO counterparts and do not appear to exhibit altitude dependent biases or photomutliplier saturation effects. This is an important result to document with regards to the long-term stability of the NDACC lidar temperature and ozone dataset at the Hohenpeißenberg Meteorological Observatory.





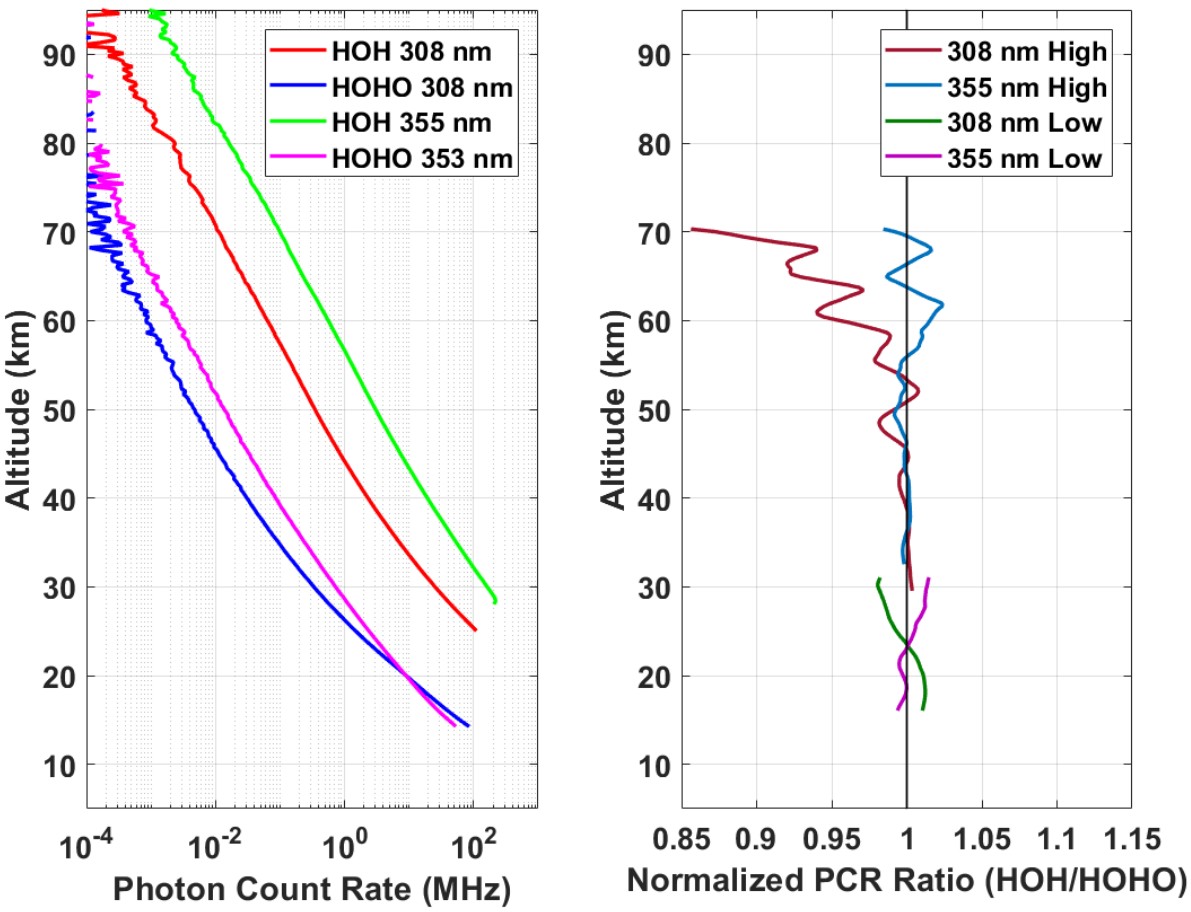

**Figure 2.** High gain signals for the HOH and HOHO lidars showing higher signal levels in the newly improved HOH lidar. Ratios of high and low gain signals for the HOH and HOHO lidars showing SNR improvement in received signal.

## 3.2 Example Comparisons of Temperature and Ozone Profiles

An example of both temperature and ozone profiles made during the HOPS campaign by each of the instruments are given in Figs. 3 and 4. Differences are taken with respect to the measured ozone and temperature from the HOH lidar when all instruments were in operation.

The temperatures shown in the left hand panel of Fig. 3 were measured on the $21^{st}$ of October 2018 and all follows the expected profile for the middle atmosphere. There is very close agreement between 30 and 50 km in the stratosphere with slightly more variation below 20 km and in the mesosphere. A closer examination of the temperature differences of each instrument with respect to the HOH lidar is shown in the right hand panel of Fig. 3. To calculate the differences, all measurements were adapted to a standard 1 km grid. Below 20 km it is expected that geophysical differences in the sampled air masses will have a larger impact on the differences with the stationary lidars than in the middle atmosphere. Additionally, the uncertainty of





the satellite measurements at low altitudes, advection of the balloon sondes, and possible contamination of the lidar signal by aerosols may all contribute to the observed differences. Above the stratopause, located near 50 km, there is again a greater chance that geophysical variability is contributing to the observed lidar-satellite differences. Above 60 km the signal to noise ratio of the HOHO lidar (cyan) becomes the largest contributor to the observed differences.

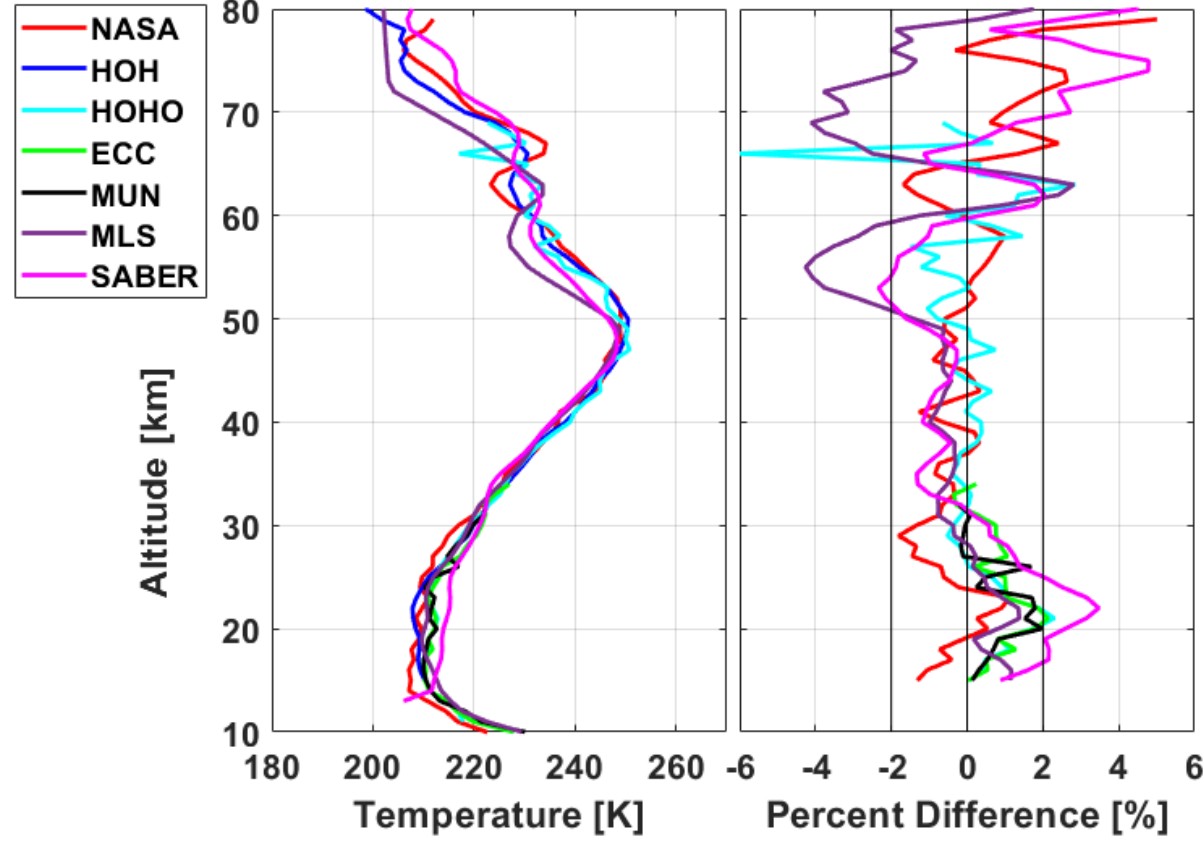

**Figure 3.** Temperature profiles measured by the NASA-STROZ (red), HOH (blue), and HOHO (cyan) lidars, locally launched ozonesonde (green), München meteorological radiosonde (black), and overpasses from MLS (violet) and SABER (magenta) on the night of 21 October 2018.

Similarly, an example of nightly average ozone profiles for the night of the $6^{th}$ April 2019 ($7^{th}$ April 2019 for OMPS) is

given in the left hand panel of Fig. 4. We can see that all instruments accurately reproduce the shape of the stratospheric ozone layer and also identify a ozone laminae near 13 km. The HOH and HOHO lidars report ozone profiles for altitudes greater than 15 km. In the right hand panel of Fig. 4 all profiles were adapted to a common 300 m grid and compared with the ozone profile measured by the HOH lidar. Below 25 km there is very good agreement between all instruments with differences of generally less than 10 %. The lowest couple of data points for the HOH lidar near 15 km may underestimate the ozone number density

on this particular night. SABER ozone was cut at 20 km as below this point the profile number densities became unrealistically





large. Above 28 km there is increased variability (expressed as percent difference) between the lidars and the satellites which is likely a function of low ozone number densities and geophysical variability. Above 40 km, the percent difference between the different measurements is not a useful metric as small absolute differences in the ozone number density can translate to very large percent differences. We will provide various other metrics later in the article when we discuss the systematic bias of

ozone measurements in this region.

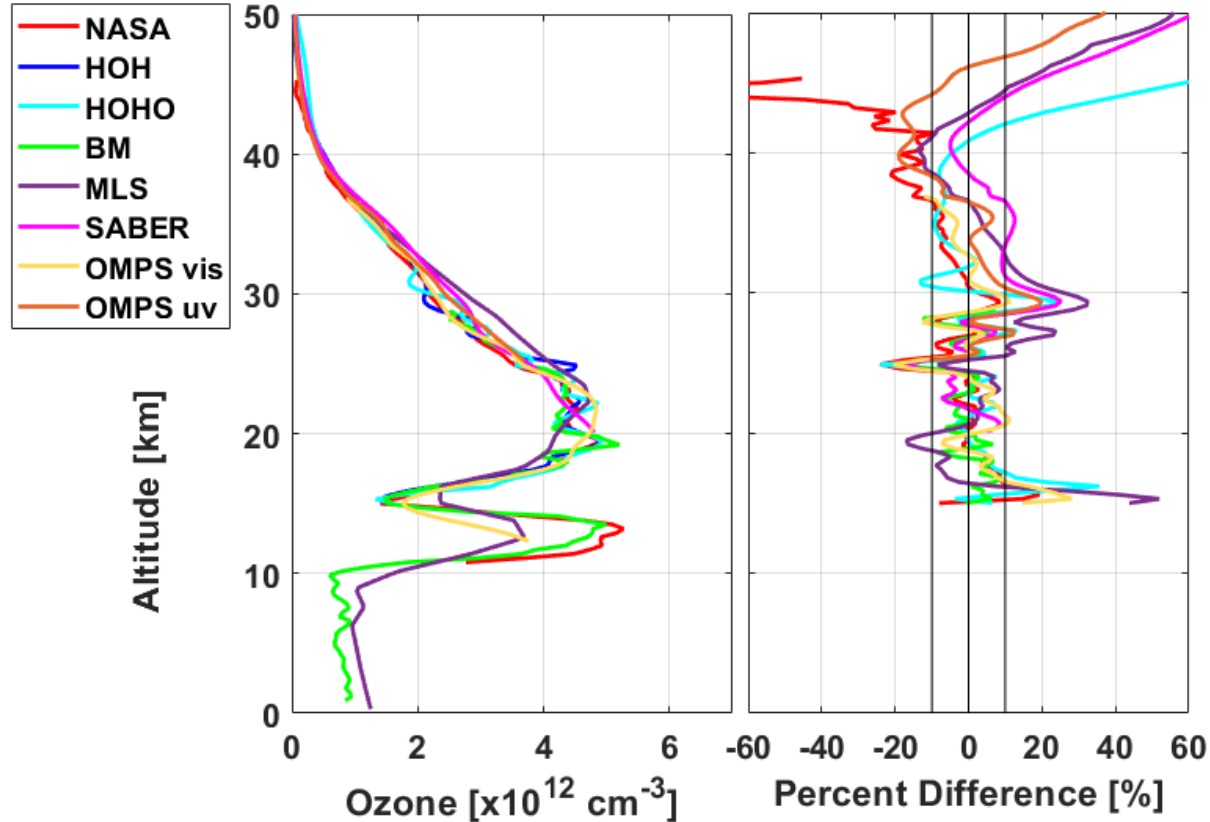

**Figure 4.** Ozone profiles measured by the NASA-STROZ (red), HOH (blue), and HOHO (cyan) lidars, locally launched ozonesonde (green), and overpasses from SABER (magenta), MLS (violet), OMPS VIS (mustard), OMPS UV (burnt orange) on the night of 6 April 2019.

## 4 Intercomparison Results for Ozone

Figure 5 shows a time series of ozone number densities measured by each of the different systems used in the HOPS campaign. The ozone profiles of each instrument were integrated to 2 km resolution before being plotted. The top panel which shows the ozone number densities at 40 km, indicates that in 2019 (last 8 nights) there was tight clustering of all the measurements while

in 2018 (first two nights) there was slightly more variation. In the second panel, which shows ozone densities at 30 km, we see that there is again tight clustering for all instruments except OMPS. This is to be expected as the OMPS given that there is a



large temporal offset (data taken the morning after the lidar measurements) and that the visible channel at 602 nm which we used for this study only extends to 35 km. The ozonesonde on the $31^{st}$ for March appears to be an outlier and is likely that well known pump problems at low pressures is the cause. The third panel at 20 km also shows very tight clustering between

all instruments with a slight high bias beginning to be seen in SABER data. The bottom panel at 15 km shows a higher level of inter-measurement variability between the lidars and satellites as the geophysical variability and sampling uncertainty become evident. SABER clearly shows a high bias with respect to other instruments at this altitude.

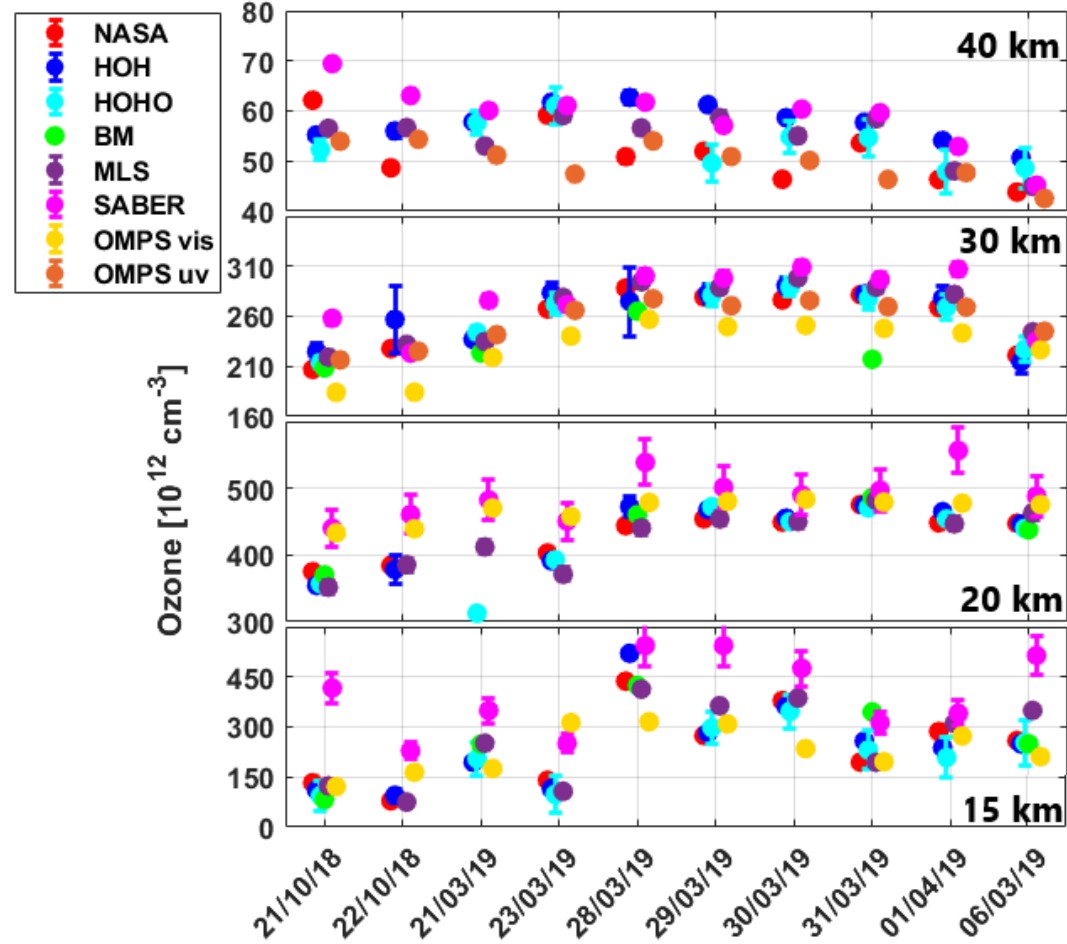

**Figure 5.** Time series of ozone measurements smoothed using a 2 km boxcar average. Top panel contains ozone densities at 40 km, second panel at 30 km, third panel at 20 km and the bottom panel at 15 km.

A more systematic look at the ensemble ozone number density differences between the HOH lidar measurements and the measurements made by each of the other instruments is shown in Fig. 6. The darkened line represents the mean difference

for each pair of measurements and the shaded region is the $2\sigma$ (95% confidence level) limit. The best agreement between the different ozone systems is found between 20 and 40 km altitude where differences are generally less than ±10%. The larger



deviation in the OMPS VIS profile (mustard) above 28 km is likely an indication that we should rely on the OMPS UV channel (burnt orange) above this height, while the sharp decrease in the Brewer-Mast profile (green) above 30 km arises from errors in the BM pump corrections at low pressure. Below 20 km there are larger differences between the satellites and the lidars (and

sondes). For MLS (violet) and OMPS VIS (mustard) this is likely due to geophysical differences in the sampled air masses while for SABER (magenta) there is a definite bias in the data. In general, the results shown in Fig. 8 are similar to the results of shown in Figure 7 of the LAVANDE study and to previous NDACC intercomparisons. Above 40 km there is an unexplained low bias of approximately 25% in the OMPS UV channel (burnt orange) and 35% in the NASA-STROZ ozone densities with respect to all other measurements. Some of the bias at this altitude may be the same as the documented low bias of 8-25 % in

OMPS UV ozone with respect to co-located profiles from OSIRIS, MLS, and ACE (Kramarova et al., 2018).



**Figure 6.** The average relative difference profile between the ozone profiles measured by the various HOPS instruments compared to the ozone profile measured by the HOH lidar. The shaded range gives the ±2 standard deviations of the mean and indicates the statistical confidence interval at the 95% uncertainty level.

Figure 7 shows the correlation, with respect to the HOH lidar, of each data point for each of the involved HOPS instruments as a function of ozone number density. In the left hand panel, showing measurements from 15 to 20 km, we see that NASA-STROZ (red), HOHO (cyan), the ozonesonde (green), and OMPS (yellow) all closely track the ozone number densities measured by the HOH lidar. The spread in of values is roughly symmetric about the reference line. MLS (violet) tends to report significantly higher values than all other instruments when measuring ozone number densities less than $3 * 10^{12} cm^{-3}$ which is consistent with the results of Figure 8 in the Wing et al. (LAVANDE, 2020) study. The central panel, showing measurements from 20 to 30 km likewise shows very little scatter for all instruments except for MLS and SABER (magenta). The increased scatted may be due to geophysical variability and fewer overall nights of comparison. This result is different than the LAVANDE study which showed very little scatter in the satellite data. The right hand panel shows the scatter for all instruments from 30 to 50 km and we can clearly see the outlier data points in the Brewer-Mast (green) where the balloon was blown away from the station.

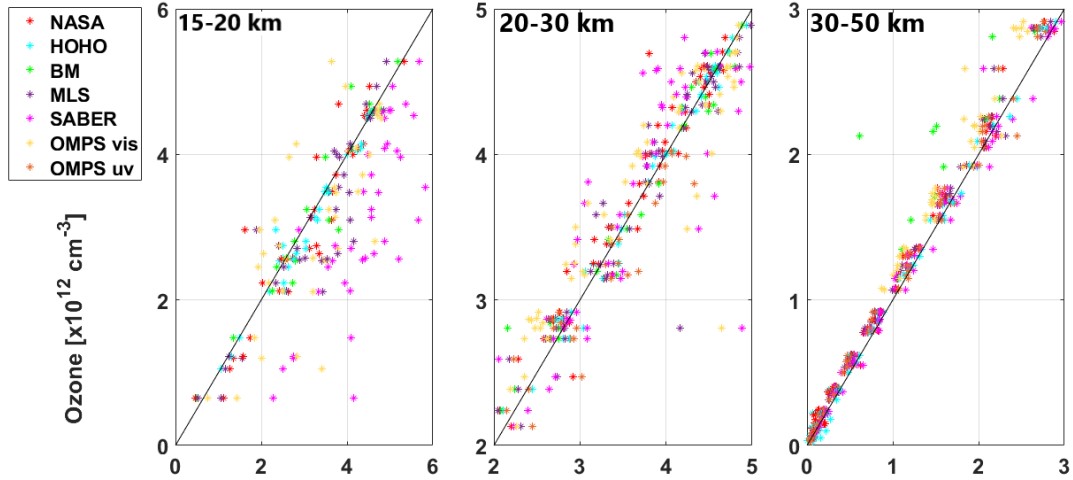

**Figure 7.** Ozone scatter plots, smoothed using a 2 km boxcar average, with respect to ozone number densities from the HOH lidar. (Left): 15 to 20 km; (Centre): 20 to 30 km; (Right): 30 to 50 km;

Examining the correlation between each of the instruments in the HOPS campaign and the HOH lidar adds another facet to our understanding of the intercomparison. By examining the 'goodness' of the match as a function of altitude (shown in Fig. 8) we can examine the difference between measurements while taking into account the statistical scatter as well as any co-variances. Unsurprisingly, the best correlation with the HOH lidar is the HOHO lidar (cyan) with correlation coefficients greater than 0.95 below 35 km. Above this altitude, the drop in the signal to noise ratio (SNR) of the HOHO lidar contributes to a rapid decrease in the correlation. In the same altitude range, MLS (violet) and the NASA-STROZ lidar (red) also show very high correlations greater than 0.85. At higher altitudes the SNR of NASA decreases the correlation to a minimum of 0.5 near 45 km while the SNR of the HOH lidar and geophysical variability gradually reduce the correlation with MLS. The correlation with the Brewer-Mast (green) is very high at low altitude and gradually descends as the pump correction becomes less reliable at lower pressures. The correlation of SABER (magenta) exhibits the 'S-shape' seen in the LAVANDE study with a dip in



the correlation values near 25 km near the maximum of the ozone concentration. This drop occurs when the instrument is not accurate enough to detect the very small changes in ozone density. The correlation profile for OMPS VIS (mustard) increases with altitude and smoothly merges with the correlation profile of the OMPS UV (burnt orange). The maximum correlation for OMPS should be read as 0.84 at 30 km as we have more confidence in the UV channel in the middle and upper stratosphere.

There are two possible explanations for why the OMPS correlation is smaller than MLS. First, there is a significant time offset between the nightly lidar measurements and the OMPS over pass which happens on the morning after. Second, as will be discussed in Fig. 14 the visible channel of OMPS has a very large estimated uncertainty below 20 km.

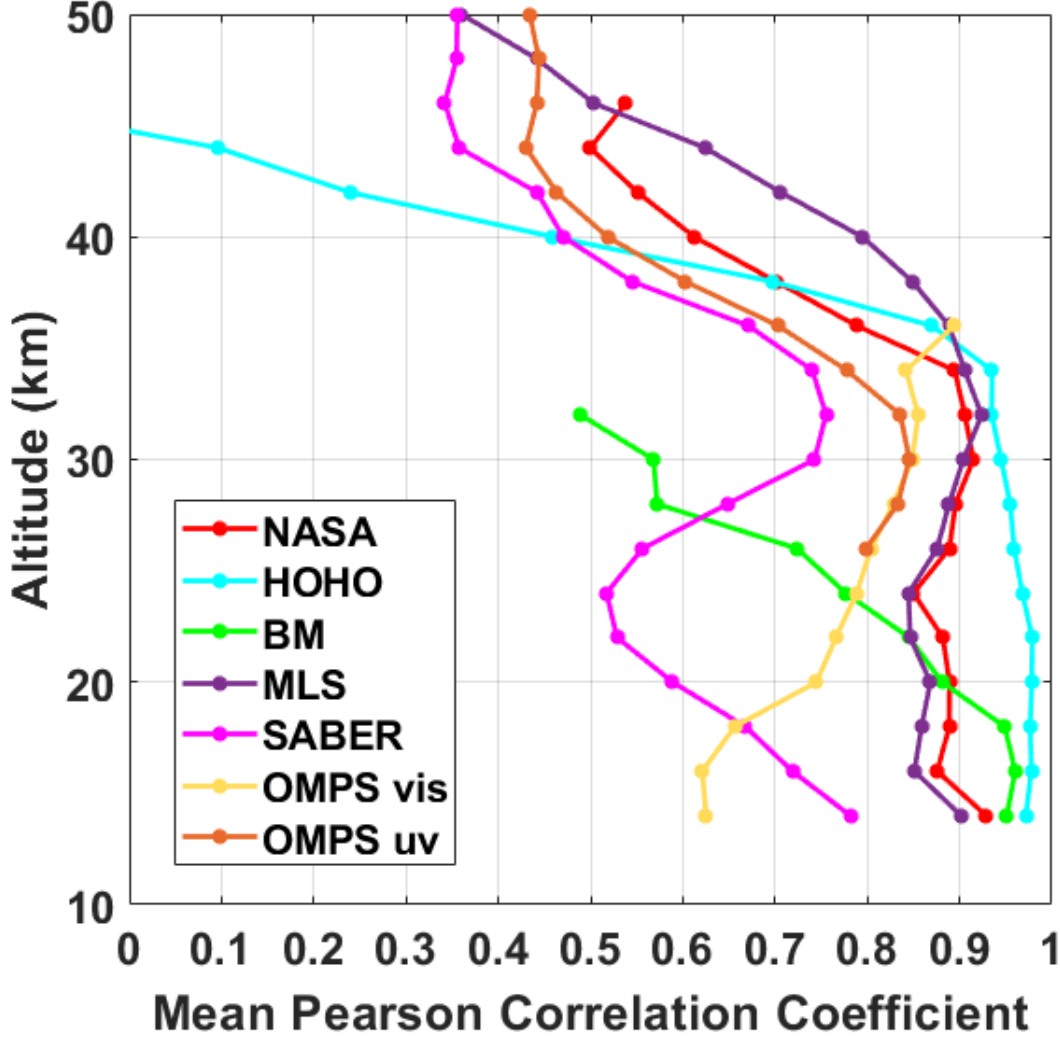

**Figure 8.** Vertical profiles of the correlation coefficient calculated for ozone concentrations measured by the various HOPS instruments with respect to HOH measurements. Correlation is taken over the 10 nights of the HOPS campaign data vertically integrated to 2 kilometres.



## 5    Intercomparison Results for Temperature

We have conducted a similar analysis for HOPS temperature measurements as was done in the previous section for ozone.
Figure 9 shows the temperature time series at four altitudes for each of the different systems during HOPS.

The top panel of Fig. 9 traces temperatures in the mesosphere at 70 km. At these altitudes we can see the contribution that larger temperature uncertainties in the lidars (particularly, HOHO (cyan)) introduces to the time series. In general, all three lidars NASA, HOH, and HOHO report higher temperatures than the satellites MLS and SABER. This result is consistent with LAVANDE as well as other European lidar-satellite comparisons (Wing et al., 2018, b).

The second panel of Fig. 9 traces temperatures at 50 km, near the altitude associated with the stratopause. With the exception of the $21^{st}$ of March 2019 where the lidars and satellites produce very different measured temperatures, the lidars and satellites generally produce similar temperatures, with the satellites being 5 K cooler than the lidars.

The third panel of Fig. 9 traces temperatures in the lower stratosphere and shows the best agreement between all measurements. We expect that the temperature at these altitudes would show very little variability due to the high SNR in all instruments as well as the low geophysical variability in lower stratospheric temperatures on hourly timescales. SABER (magenta) appears as an outlier from the $31^{st}$ of March to the $6^{th}$ of April 2019. The average spatio-temporal offset of the SABER profiles from Hohenpeißenberg Meteorological Observatory is not significantly different from the $21^{st}$ to the $30^{th}$ of March 2019.

The bottom panel of Fig. 9 traces temperatures at 10 km in the UTLS. At this altitude there are fewer measurements, more geophysical variability associated with passing weather fronts, variability associated with the advection of balloon measurements, and possible bias introduced into the lidar temperatures from aerosol contamination. Despite the increased variability, we can see that the NASA lidar (red), SABER (magenta) and MLS (violet) generally measure colder temperatures than the meteorological radiosonde from the München station (black) and the locally launched Brewer-Mast sonde (green). It is interesting to note that the two balloon sonde measurements agree very well despite the 65 km separation between Hohenpeißenberg and München.



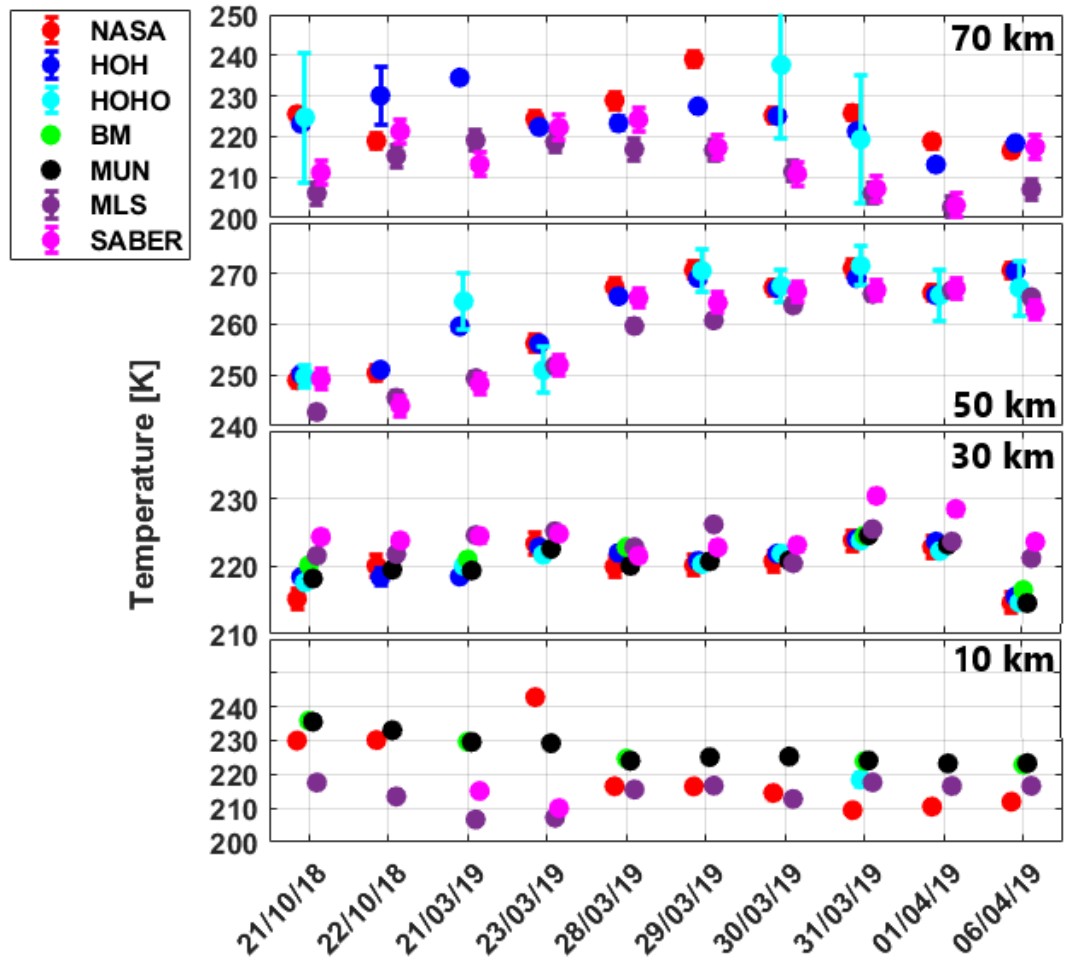

**Figure 9.** Time series of temperature measurements at smoothed using a 2 km boxcar average. Top panel contains temperatures at 70 km, second panel at 50 km, third panel at 30 km and the bottom panel at 10 km.

The average temperature difference between all instruments participating in the HOPS campaign and the NASA lidar is given in Fig. 10. The differences between the temperatures produced by the three lidars are less than ±5 K from 15 to 80 km. The temperature differences between HOH and NASA (red) are only significant below 18 km and above 78 km. The NASA temperatures appear to have a slight cold bias below 30 km which is consistent with the results from LAVANDE described in the introduction Sect. 1.2. MLS (violet) becomes significantly different from the other measurements above 55 km and

exhibits a vertically oscillating temperature bias described in (Wing et al., 2018b). SABER (magenta) exhibits a significant warm bias between 15 and 25 km with respect to lidar measurements which has been previously been identified in the SABER temperature assessment paper (Remsberg et al., 2008).







**Figure 10.** Average absolute differences with respect to the NASA temperature profile measured during the HOPS campaign. The shaded range gives ±2 standard deviations of the mean, and indicates statistical uncertainty at the 95% confidence level.

Figure 11 shows the scatter between nightly temperature comparisons during the HOPS in three panels. The left hand panel shows the differences between temperatures from each instrument and the HOH lidar temperatures in the UTLS from 10 to





35 km. The scatter shows fairly close agreement to the black 1:1 reference line, particularly for the in-situ temperatures from the sondes. NASA (red) exhibiting slightly colder temperatures and the satellites MLS (violet) and SABER (magenta) having slightly warmer temperatures. The centre panel shows the scatter from 35 to 60 km in the upper stratosphere and stratopause region. The temperatures from NASA fall very closely along the reference line however, the temperatures from HOHO (cyan) exhibit more variation associated with the drop in SNR in that system. The satellites have a very high level of temperature

variation but appear to be centred about the reference line. In the right hand panel the temperature scatter from 60 to 90 km is shown. The temperature variance is largest at these altitudes however, despite the increased scatter we can see the systematic cool bias of the satellites and warm bias of NASA with respect to HOH.

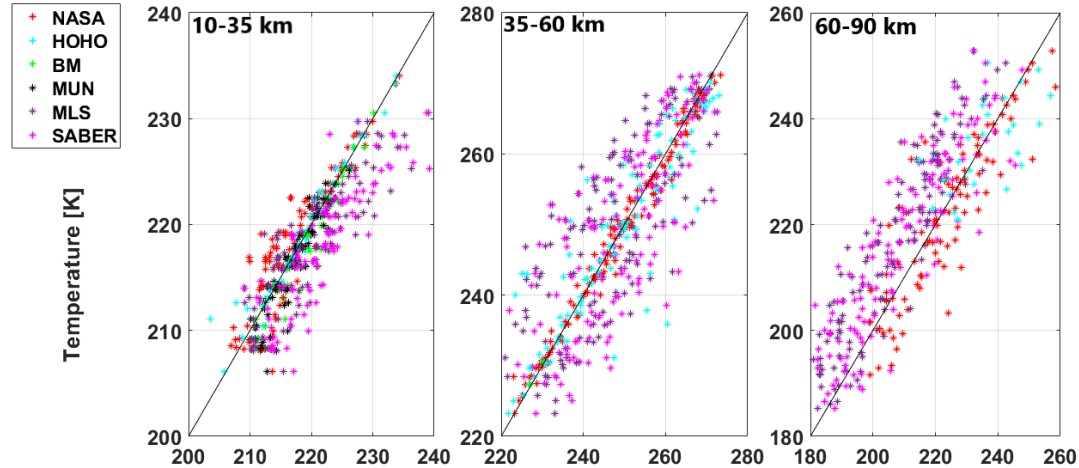

**Figure 11.** Scatter plots of temperature at 2 km resolution for the different instruments involved in the HOPS campaign with respect to the HOH lidar. (Left): 10 to 35 km; (Centre): 35 to 60 km; (Right): 60 to 90 km;

Figure 12 shows profiles of the mean Pearson's correlation coefficient of HOPS instrument temperatures with respect to temperatures from the HOH lidar, similar as to what was done for ozone. Below 32 km we see very high correlation between

the two balloon sondes and the HOHO lidar all three of which show correlations larger than 0.85. The HOHO lidar reaches a maximum correlation with HOH of 0.98 near 35 which slowly declines with altitude as the SNR of HOHO approaches 1. The correlation profile of the NASA lidar temperatures is unique in that there is lower correlation from 15 to 30 km. This drop may be due to the combination of HOH lidar data with the radiosonde mentioned in Sect. 2.2, differences in the overlap correction between the two lidar systems, the use of Raman temperature channels in the NASA-STROZ lidar, or geophysical sampling

problems arising from a few nights where the DWD lidars measured longer than the NASA system. Disentangling the source of the disagreement is beyond the scope of a 'blind intercomparison' and would require each team to reprocess their data. In Sect. 6 we will discuss the disagreement between the observed differences between NASA and HOH and the differences that we should expect given the reported uncertainty budgets of each system. From 35 to 58 km the correlation between NASA and HOH temperatures is very high at nearly 0.99. Above 60 km the statistical variation and the differences in filtering could





reduce the value of the correlation. Both MLS and SABER have very similar correlation profiles with respect to the HOH lidar temperatures. There is a local minimum in both profiles near 30 km which is associated with the low geophysical variance between different measurements and even lower reported uncertainties at the start of the stratosphere. The general drop in correlation coefficients for all measures above 50 to 60 km can be attributed to the increase in both the measurement variance and the drop in SNR for the lidar systems.



**Figure 12.** Vertical profiles of the correlation between temperatures reported by the various HOPS systems with respect to the HOH lidar temperature profiles.





## 6 Intercomparison of Uncertainty Estimates

When comparing the published uncertainty estimates for lidars, sondes, and satellites during an intercomparison it is not sufficient to rely simply on the reported instrument precision. Some instruments report full uncertainty budgets, others average accuracy, and others single profile precision. To make the comparison fair we have taken the average of the total nightly uncertainty for each instrument and normalised it with respect to the nightly average measurement to arrive at a plot estimating the average relative uncertainty as a function of altitude during HOPS. This follows the same method used in LAVANDE.

For the lidars the largest terms in the uncertainty budgets are the statistical uncertainty arising from the Poisson counting statistics for photon detection which become large at higher altitudes. Several other smaller corrections with respect to atmospheric scattering and transmission, instrument corrections, and algorithm initialization (temperature only) are also included in the formal 'NDACC' uncertainty budget described in details in Leblanc et al. (2016a, b, c). In this blind intercomparison we take the reported total uncertainty or 'NDACC' uncertainty reported for by each group. In Fig. 13 we can see that the average of the nightly relative uncertainty for temperature in the NASA (red), HOH (blue), and HOHO (cyan) lidars are typically less than 1 % over most of the measurement range. The HOHO lidar which has a less powerful laser output at 353 nm (refer to the introduction Sect. 3.1 and the discussion of Fig. 2) reaches 1% relative uncertainty at 45 km – much lower than the HOH and NASA lidars. The sudden drop in the relative uncertainty in all three lidar profiles near 25 km is associated with the transition from the low gain lidar channels to the high gain lidar channels. The nightly uncertainty profiles for MLS (violet) and SABER (magenta) were downloaded directly with the temperature profiles. Temperature uncertainties for the Vaisala RS41-SGP radiosonde used at both the München radiosonde station and the Vaisala RS92-SGP attached to the Brewer-Mast launched at Hohenpeißenberg is given as 0.15 K below 100 hPa and 0.3 K above 100 hPa. We have not included these values in Fig. 13 as they are too small to be clearly distinguishable.

**Figure 13.** Relative uncertainties in temperature for all HOPS instruments.

The major term in the uncertainty budget for the lidar ozone measurements comes from the Poisson photon counting uncer-
tainty. A full and detailed propagation of uncertainty through the lidar equation is given in Godin et al. (1999). In Fig. 14 we





see different behaviours in the relative uncertainty of the NASA (red) and the uncertainties and the HOH (blue) and HOHO (cyan) lidars. The peak in relative uncertainty between 25 and 30 km in both DWD lidars is due to the transition between the low gain and high gain lidar channels. It is recommended that the DWD lidars merge their high and low gain channels at a

lower altitude to suppress the uncertainty peak in this range. The ozone uncertainty in the Brewer-Mast is given simply as $\pm$3-5%. This flat uncertainty profile does not capture the observed variance between the Brewer-Mast measurement and the lidars which is discussed in the next section. We have chosen to include an uncertainty profile estimated for the ECC (green) by Tarasick et al. (2016) which presents a more realistic uncertainty profile for a similar instrument. The relative uncertainty profile for MLS (violet) and SABER (magenta) were calculated using the uncertainty information included in the downloaded

data files. MLS has low relative uncertainty throughout most of the stratosphere, averaging 2 to 3%. The uncertainty rapidly increases at low ozone densities below 20 km and above 45 km. SABER ozone uncertainty appears unrealistic above 35 km and increases rapidly below 30 km. We have endeavoured to estimate accurate profiles of OMPS relative uncertainty for both the visible channel and the UV channel for the HOPS campaign. Using the $1\sigma$ measurement uncertainty estimates found in Loughman et al. (2005); Zawada et al. (2018); Kramarova et al. (2018) we have calculated the nightly relative uncertainty

profiles for OMPS visible (mustard) and UV channels (burnt orange). We have doubled the $1\sigma$ values and then averaged the nightly relative uncertainty profile for HOPS to generate an uncertainty profile which is consistent with the other participating measurements at $2\sigma$. This value is approximately 3% for the OMPS visible channel between 20 and 30 km. The relative uncertainty rises drastically below 18 km and increases slightly above 32 km. Likewise, the relative uncertainty profile for the UV channel of OMPS uses the reported $1\sigma$ precision and accuracy estimates by Loughman et al. (2005) of between 1 and 3% to

calculate the relative uncertainty profile in Fig. 14 (burnt orange). The rapid increase in relative uncertainty seen in the OMPS UV channel above 35 km results from the rapid decrease in ozone number density and the possible low bias of OMPS UV data seen in Fig. 6.





**Figure 14.** Relative uncertainties in ozone for all HOPS instruments.

## 6.1 Assessment of the Uncertainties Reported by the Instruments

Here we conduct an intercomparison of the reported uncertainty budgets for all HOPS lidars for both temperature and ozone.
This exercise is important for establishing that the total uncertainty budgets for NDACC lidars are realistic and in keeping
with NDACC guidelines and standards. For lidar-lidar comparisons there is nearly perfect spatio-temporal coincidence and we
can neglect geophysical variations in our uncertainty comparison. Here we will use the NASA-STROZ (red) average relative





uncertainty profile as the reference. Following the same statistical comparison technique used in the companion Wing et al. (2020) article we will assume that there is no correlation between the average measurement noise for the lidars. In Fig. 15

the measurement uncertainty of NASA-STROZ lidar, $\sigma_N$ (red), HOH lidar, $\sigma_H$ (blue), HOHO lidar, $\sigma_{Ho}$ (cyan), are plotted alongside the combined uncertainty, $\sigma_{combined}$ (black), given in 2, and the relative standard deviation of the measurement differences, $\sigma_{RSD}$ (grey), given in 1. In these equations, $N_i$, describes the NASA measurement, $\overline{N}$, described the average NASA measurement, $\sigma_N$, describes the measurement uncertainty for NASA, $X_i$, $\overline{X}$, and $\sigma_X$, describe same properties for the HOPS instrument under consideration, and n is the total number of measurements.

$$\sigma_{RSD} = \sqrt{\left(\frac{1}{n-1}\right)\Sigma\left(\left(\frac{X_i}{N_i}\right) - \left(\frac{\overline{X}}{\overline{N}}\right)\right)^2} \tag{1}$$

If the combined uncertainty estimates, expressed in Eq. 2 (black) are correct, they should be similar to the observed standard deviation of all the nightly mean ozone profile differences, $\sigma_{RSD}$ (grey), expressed in Eq. 1.

$$\sigma_{combined} = \frac{\overline{X}}{\overline{N}}\sqrt{\left(\frac{\sigma_X}{\overline{X}}\right)^2 + \left(\frac{\sigma_N}{\overline{N}}\right)^2} \tag{2}$$

Figure 15 compares the average relative uncertainties for the three lidars participating in the HOPS campaign for both

temperature and ozone. In panel a) we see the comparison of the relative temperature uncertainty for the NASA (red) and HOH (blue) lidars. Above 35 km the combined uncertainty budget (black) is dominated by NASA which has a smaller receiver telescope than the HOH lidar (see Tab. 4) which results in a reduced photon count rate at higher altitudes. Below 35 km, the HOH lidar has the larger contribution to the combined relative uncertainty budget arising from increased measurement uncertainty in the low gain 355 nm channel. When comparing the combined estimated relative uncertainty (black) with the

observed standard deviation (grey) we see that below 55 km there is variance between the lidar temperature measurements which cannot be explained by the combined uncertainty budget. A nearly identical result was found in the LAVANDE study with unexplained variance below 55 km between NASA and the OHP temperature lidar, LTA, and between NASA and the OHP stratospheric ozone lidar, LiO$_3$S.

In panel b), the uncertainty of the HOHO lidar (cyan) is the largest contributor to the combined estimated relative uncertainty

budget (black). The combined uncertainty accounts for most of the observed variance in the comparison with NASA (grey) except for a discrepancy between 25 and 35 km. This region appears to be directly above the transition from the low gain to high gain channels in the HOHO lidar and the estimation of the HOHO uncertainty in this region may not be complete. Taken together with our interpretations of the LAVANDE results and the results shown in 15a) we begin to see a pattern of increased variability between lidar measurements in the region surrounding the transition between high and low gain channels which is

not fully accounted for the the NDACC uncertainty budget.

Panel c) shows the relative uncertainty in ozone for NASA (red), HOH (blue), combined uncertainty (black), and observed variation between measurements (grey). As was previously stated, the differences between the estimated relative uncertainty profiles for the NASA and HOH lidars arises from thre transition from the low gain to high gain channels in the DWD lidras.



The observed variance is well represented by the combined uncertainty above 23 km. From 15 to 23 km there is more variation
in the data than can be accounted for in the uncertainty estimates of either lidar. One possible explanation for the increased
variability below 25 km is sampling time. On a few nights the NASA-STROZ lidar measured for a set number of hours
while the DWD lidars measured for the entire night. Given that this is a 'blind intercomparison' we cannot reprocess the data
however, in future NDACC validation exercises we strongly encourage participating PIs to end measurements at the same time
or submit partial files to the NDACC referee. Below 25 km there is sufficient geophysical variation that a few hours of extra
measurements can change the nightly mean profile.

Panel d) shows the ozone relative uncertainty estimates from NASA (red) and HOHO (cyan). Similar to the results in panel
b), the combined uncertainty and observed standard deviation are dominated by the uncertainty estimates of the less powerful
HOHO lidar.





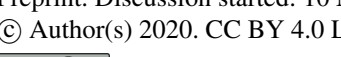

**Figure 15.** Comparison of relative uncertainties in both temperature and ozone for the NASA (red), HOH (blue), and HOHO (cyan) lidar systems.





## 6.2 Uncertainty Evaluation of the satellites

In the LAVANDE companion paper we attempted to separate the measurement uncertainty associated with each profile taken during a satellite overpass, the sampling uncertainty associated with the variation between individual profiles included in the average satellite overpass, and the geophysical variability. It was correctly pointed out that characterisations of sampling uncertainty are not completely independent of geophysical variability. For the HOPS intercomparison of lidar-satellite relative uncertainty estimates we have not attempted to address sampling uncertainty. In all cases where the observed standard deviation

of the differences between observations (grey) is larger than the combined NASA-satellite estimated uncertainty budget (black) we will interpret the difference as 'geophysical variability' with the understanding that there is some unknown contribution associated with the accuracy of the satellite measurement.

In Fig. 16 a) and b) we can see that below approximately 70 km the combined uncertainty budget is mostly due to the contributions of the MLS (violet) and SABER (magenta) measurement uncertainties respectively. Above this altitude, the

statistical measurement uncertainty in the lidar temperature measurements become larger than the measurement uncertainty in the satellites

Figure 16 panels c), d), e), and f) show the relative uncertainty estimates for ozone for NASA (red) with MLS (violet), SABER (magenta), OMPS visible channel (mustard) and OMPS UV channel (burnt orange), respectively. For MLS and SABER comparisons the satellite measurement uncertainty estimates are larger than the lidar uncertainty estimates below

30 to 35 km, with the opposite holding true at higher altitudes. In panel e) the measurement uncertainty in OMPS VIS (mustard) is much larger than the measurement uncertainty in the lidar and accounts for nearly all the observed variation (grey). The OMPS UV measurement uncertainty (burnt orange) shown in panel f), is comparable to the lidar measurement uncertainty above 45 km and is larger than the lidar measurement uncertainty at lower altitudes.





**Figure 16.** Comparison of relative uncertainties in both temperature and ozone for the NASA lidar (red), MLS (violet), SABER (magenta), OMPS visible (mustard), and OMPS UV (burnt orange) .





### 6.3 Uncertainty Evaluation of the balloon sondes

The temperature measurement uncertainty for NASA lidar (red) and the balloon borne in-situ measurements (green), shown in Fig. 17 a) are very similar in magnitude. However, the combined uncertainty (black) is consistently less than the observed standard deviation between the lidar and sonde temperature measurements (grey). We expect that the majority of the difference observed variation below 20 km is likely due to the geophysical variability inherent in sampling different air masses while variability in ozone above 27 km likely arises from problems with the ozonesonde pump or poor pump corrections. Figure

17 b) shows that the combined ozone uncertainty (black) is an overestimates the observed standard deviation between 22 and 26 km and severely underestimates the variation at lower altitudes. Recall that we have used the ECC uncertainty reported by Tarasick et al. (2016) in place of the reported 3 to 5% uncertainty associated with the Brewer-Mast. Given that the ECC uncertainty is 2.5% at 25 km and 4% at 15 km, the generic Brewer-Mast uncertainty profile would further overestimate the combined uncertainty above 22 km. A better estimation of measurement uncertainty for the Brewer-Mast instrument is required

for the HOPS campaign.

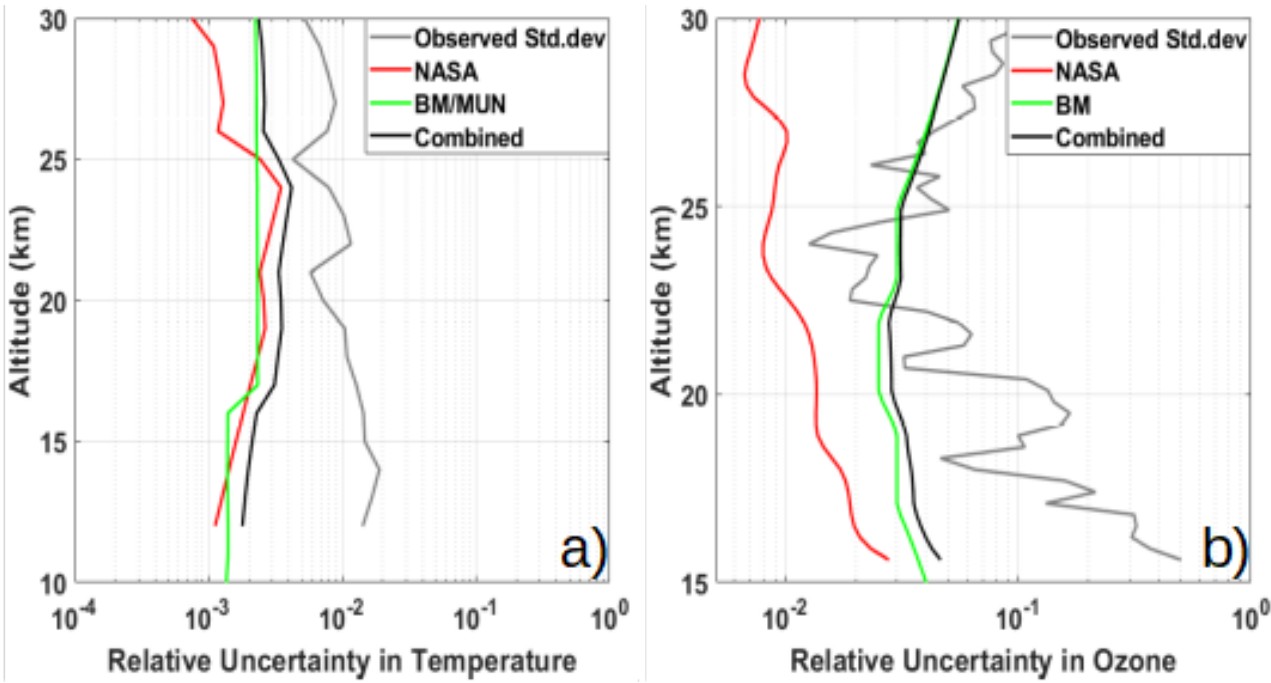

**Figure 17.** Comparison of relative uncertainties in both temperature and ozone for the RS41-SGP/RS92-SGP and ECC sondes.

### 7 Cross-Intercomparison with results obtained during the LAVANDE campaign

Since 2018 substantial work has been done to validate European NDACC lidar activities using the travelling standard NASA-STROZ lidar. Given that the LAVANDE and HOPS campaigns were conducted one after another we are presented with the





opportunity to conduct a cross-intercomparison analysis of the NASA-STROZ lidar as well as make some comments on how
the five NDACC stratospheric lidars (NASA, LiO$_3$S, LTA, HOH, and HOHO) have compared against the satellite and sonde
measurements.

Figure 18 shows the average percent difference in ozone profiles for all instruments during both the LAVANDE and HOPS
intercomparison and validation campaigns (except the tropospheric ozone lidar at OHP, LiO$_3$T). Panel a) shows the average
differences between NASA and LiO$_3$S (blue), HOH (red), and HOHO (cyan). Above 40 km NASA consistently reports lower
ozone densities than the three other stratospheric ozone lidars. Below 20 km NASA reports higher ozone densities than the
DWD lidars (HOH and HOHO) but lower densities than LiO$_3$S at OHP. Between 20 and 32 km all lidars report statistically
identical ozone number densities.

Panel b) shows the average ozone differences between the NASA lidar and the ozonesonde measurements is shown. The
ECC (green) was flown during the LAVANDE campaign at OHP while the Brewer-Mast (red) was flown during the HOPS
campaign at Hohenpeißenberg. Below 20 km NASA reports lower ozone densities than the ozonesondes however, this is only
significant below 15 km for the ECC. During the LAVANDE campaign the ozonesondes were in significantly better agreement
with the OHP lidar LiO$_3$S than with the NASA lidar below 17 km and similarly during the HOPS campaign the ozonesondes
were in better agreement with the HOH and HOHO lidars than with the NASA lidar however, this result is not significant.
Above 30 km the ozonesondes diverge from both the lidar and satellite measurements during both campaigns.

Panel c) shows the average ozone differences between the NASA lidar and SABER during LAVANDE (blue) and HOPS
(red). Above 40 to 45 km SABER ozone measurements report higher ozone densities than any of the NDACC lidars. As well
below 25 km SABER ozone profiles become undependable.

Panel d) shows the average ozone differences between the NASA lidar and MLS during LAVANDE (magenta) and HOPS
(green). Above 40 to 45 km MLS ozone measurements report higher ozone densities than any of the NDACC lidars. Below
15 to 17 km NASA and LiO$_3$S report lower ozone number densities than MLS while HOH and HOHO report slightly higher
ozone densities.





**Figure 18.** Cross-comparison of LAVANDE and HOPS ozone measurements for the NDACC lidars LiO3S, HOH, and HOHO; satellites MLS and SABER; and ozonesondes Brewer-Mast and ECC; with respect to the NASA-STROZ lidar.

Figure 19 shows the average absolute temperature difference between all instruments during both the LAVANDE and HOPS campaigns. Panel a) shows the average differences between NASA and LiO$_3$S (blue), LTA (green), HOH (red), and HOHO (cyan). Above approximately 60 km NASA consistently reports warmer temperatures than three of the other other temperature

lidars (LiO$_3$S truncates temperature profiles at 60 km). This warm bias is significant with respect to the OHP temperature lidar





LTA above 65 km. Below 20 km the NASA lidar reports colder temperatures than both the LiO$_3$S and HOH lidars. As was previously reported in Wing et al. (2020), the OHP temperature lidar was discovered to have a faulty photomultiplier tube in the low gain channel (0 to 50 km) which accounts for the warm bias between 30 and 50 km. The LAVANDE study allowed us to identify and replace this faulty component and the subsequent temperatures no longer show this bias. LTA temperatures

below 30 km have an aerosol induced cold bias which is more pronounced for temperature lidars operating at 532 nm than for lidars at 355 nm.

Panel b) shows the average temperature differences between the NASA lidar and the sonde measurements. The temperatures from the ECC which was flown with MeteoModem M10 radiosondes (green) and Brewer-Mast flown on a Vaisala RS92-SGP (red) are shown. At all altitudes the NASA lidar has a slight cold bias of between 2 and 4 K.

Panel c) shows the average temperature differences between the NASA lidar and SABER during LAVANDE (blue) and HOPS (red). There appears to be a general trend where NASA temperatures have a 0 to 5 K cold bias below 40 km which increases almost linearly to a slight (but not significant) warm bias near 80 km.

Panel d) shows the average temperature differences between the NASA lidar and MLS during LAVANDE (magenta) and HOPS (green). Again the slope of the temperature difference curves is consistent between both campaigns. There is a slight

NASA cold bias near 15 km which increases to a slight NASA warm bias near 70 km. The large, coherent, oscillating structures present at 50 and 60 km are known to result from errors in MLS geopotential (Wing et al., 2018b).





**Figure 19.** Cross-comparison of LAVANDE and HOPS temperature measurements for the NDACC lidars LiO3S, LTA, HOH, and HOHO; satellites MLS and SABER; and the radiosondes attached to the Brewer-Mast (RS92-SGP) and ECC (MeteoModem M10); with respect to the NASA-STROZ lidar.

In the introduction we gave the NDACC standard for ozone lidars as having an accuracy better than ± 3% between 12 and 35 km and an accuracy of better than ± 10% between 35 and 40 km. The accuracy for NDACC temperature lidars was given





as agreement better than $\pm$ 1 K. Table 1 displays a summary of where each of the NDACC lidars is statistically equal at the $2\sigma$
(95% confidence level) to NASA-STROZ at the given accuracy threshold.

| Altitude (km) | Ozone at 3% | | Ozone at 10% | | Temp. at 1 K | |
|---|---|---|---|---|---|---|
| | z_min | z_max | z_min | z_max | z_min | z_max |
| LiO$_3$S | 17 | 40 | 10 | 44 | 22 | 60$^a$ |
| LTA | - | - | - | - | 50 | 68 |
| LiO$_3$T | 13 | 14.5 | 13 | 14.5$^b$ | - | - |
| HOH | 17 | 41 | 15 | 41 | 17 | 78 |
| HOHO | 16.5 | 43 | 10 | 44 | 18 | 70 |

**Table 1.** Summary of the altitude ranges over which participating OHP and DWD lidars meet NDACC accuracy requirements at the $2\sigma$ confidence level with respect to the travelling standard NASA-STROZ lidar. $^a$LiO$_3$S only reported temperatures up to 60 km during LAVANDE. $^b$LiO$_3$T is a tropospheric system and has minimal overlap with the stratospheric lidars.

## 8   Conclusions

The HOPS intercomparison campaign of the DWD lidars at the Hohenpeißenberg Meteorological Observatory with the travelling standard NDACC reference lidar NASA-STROZ has demonstrated the consistency of the HOH lidar measurements with respect to measurements made using the HOHO lidar. We have confidence in the continued high quality of the Hohenpeißen-
berg dataset for both temperature and ozone after the installation of a new lidar system.

The intercomparison exercise has confirmed that the original DWD lidar, HOHO continues to meet NDACC standards for ozone profiles at the 3% level between 16.5 and 43 km and at the 10% level between 10 and 44 km. The HOHO lidar meets the NDACC temperature standards for accuracy at the $\pm 1$ K level between 18 and 70 km. The new DWD lidar, HOH, meets the 3% ozone standard between 17 and 41 km, the 10% ozone standard between 15 and 41 km, and the $\pm 1$ K temperature standard
between 17 and 78 km.

The cross-comparison of NDACC campaign at Hohenpeißenberg Meteorological Observatory (HOPS) and at Observatoire de Haute Provence (LAVANDE) has allowed for the unique opportunity to assess potential biases in the NASA-STROZ reference lidar. When cross-compared against the LiO$_3$S, LTA, and HOH lidar temperature profiles, and MLS and SABER satellite temperature profiles, the NASA-STROZ lidar appears to have a warm bias above 60 km. The NASA temperatures have an
apparent cold bias below 30 km when cross-compared to all other instruments. These possible biases may arise from algorithm initialisation choices and serve as strong motivation for another NDACC temperature algorithm paper.

When the ozone density profiles are cross-compared for both HOPS and LAVANDE instruments there is a high degree of variability in all of the stratospheric lidars below 20 km. The NASA lidar measures higher ozone densities than the DWD lidars but lower densities than the OHP lidar. At altitudes above 40 km, the NASA lidar and OMPS-LP UV measure lower ozone
density than LiO$_3$S, HOH, HOHO, MLS, and SABER.



*Data availability.* The data that support the findings of this study are openly available [1] The data used in this publication were obtained from Hohenpeißenberg Meteorological Observatory as part of the Network for the Detection of Atmospheric Composition Change (NDACC) and are publicly available ftp://ftp.cpc.ncep.noaa.gov/ndacc/station/hohenpei/, last access: 15 March 2020 [2] local radiosondings from München http://weather.uwyo.edu/upperair/sounding.html, last access: 15 March 2020 [3] MLS temperature and ozone profiles https://disc.gsfc.nasa.
gov/datasets?keywords=MLS, last access: 15 March 2020 [4] SABER temperature and ozone profiles http://saber.gats-inc.com/, last access: 15 March 2020, and [5] The OMPS LP version 2.5 ozone profiles https://doi.org/10.5067/X1Q9VA07QDS7, last access:3 August 2020

*Author contributions.* TJM, JTS, GS, and WS conducted the measurement campaign at Hohenpeißenberg. SGB conducted the blind comparison of all HOPS data. RW drafted the article. TJM, JTS, and WS provided access to the data and instruments. SK processed the OMPS data. All authors discussed the results and contributed to the final paper.

*Competing interests.* The authors declare that they have no conflict of interest.

*Acknowledgements.* This work is supported by the Deutscher Wetterdienst (DWD), the Institut National des Sciences de l'Univers/Centre National de la Recherche Scientifique (INSU/CNRS), and the NASA Upper Atmospheric Research Program.





| Date | NASA | HOH | HOHO | BM | MUN | SABER | MLS | OMPS |
|------|------|-----|------|-----|-----|-------|-----|------|
| 18/10/21 | X | X | X | X | X | X | X | X |
| 18/10/22 | X | X | - | - | X | X | X | X |
| 19/03/21 | - | X | X | X | X | X | X | X |
| 19/03/23 | X | X | X | - | X | X | X | X |
| 19/03/28 | X | X | - | X | X | X | X | X |
| 19/03/29 | X | X | X | - | X | X | X | X |
| 19/03/30 | X | X | X | - | X | X | X | X |
| 19/03/31 | X | X | X | X | X | X | X | X |
| 19/04/01 | X | X | X | - | X | X | X | X |
| 19/04/06 | X | X | X | X | X | X | X | X |
| total | 9 | 10 | 8 | 5 | 10 | 10 | 10 | 10 |

**Table 2.** Measurement dates for all instruments during the HOPS campaign in October 2018 and March 2019. The dates are taken at the UT start time of the lidar measurements. X denotes a valid measurement for the given night.





**Table 3.** Instruments compared during the HOPS campaign in October 2018 and March/April 2019.

| Instrument | Measurement of ozone | Altitude range | Measurement of temperature | Altitude range | Data source |
|---|---|---|---|---|---|
| NASA-STROZ | DIAL (308 and 355 nm) | 10 to 50 km | Rayleigh and Raman lidar (355 nm) | 10 to 80 km | [1] |
| HOH | DIAL (308 and 355 nm) | 15 to 60 km | Rayleigh and Raman lidar (355 nm) | 15 to 90 km | [1] |
| HOHO | DIAL (308 and 353 nm) | 15 to 50 km | Rayleigh and Raman lidar (353 nm) | 15 to 70 km | [1] |
| Brewer-Mast sondes | KI electro chemical cell | 0 to 35 km | Platinum Resistor (RS92-SGP) | 0 to 35 km | [1] |
| München radiosondes | - | - | Platinum Resistor (RS41-SGP) | 0 to 35 km | [2] |
| MLS satellite, Version 4.23 | $\mu$wave limb sounding (240 GHz) | 10 to 80 km | $\mu$wave limb sounding (118 GHz) | 15 to 90 km | [3] |
| SABER satellite, Version 2.0 | IR limb sounding (9.6, 1.27 $\mu$m) | 15 to 90 km | IR limb sounding (4.3, 15 $\mu$m) | 10 to 100 km | [4] |
| OMPS-LP satellite, Version 2.5 | Visible and UV limb sounding | 10 to 60 km | - | - | [5] |



**Table 4.** Technical specifications for the lidars participating in the HOPS campaign.

|  | NASA | HOHO | HOH |
|---|---|---|---|
| **Transmitter** | | | |
| $\lambda_{on}/\lambda_{off}$ | 308/355 nm | 308/353 nm | 308/355 nm |
| laser @ $\lambda_{on}$ | Light Machinery IPEX 868 | Lambda Physik LPX 220i | Coherent LPX 210i |
| laser @ $\lambda_{off}$ | Continuum 9050 | $H_2$ Raman Cell | Innolas Spitlight 600 |
| pulse energy @ $\lambda_{on}/\lambda_{off}$ | 300/150 mJ | 150/15 mJ | 200/120 mJ |
| laser rep. rate | 100/50 Hz | 35 Hz | 20 Hz |
| **Receiver** | | | |
| telescope | Dall-Kirkham | Newtonian | Newtonian |
| mirror diameter | 0.76 m | 0.6 m | 1.0 m |
| field of view | 2.3 mrad | 0.4 mrad | 2 mrad |
| focal length | 3.66 m | 2.4 m | 3 m |
| parallax | 0.75 m | 0.7 m | 0.8 m |
| high & low gain channels | 308, 355, 387 nm | - | 308, 355 nm |
| single channels | 332, 407 nm | 308, 353 nm | 332, 387 nm |
| **Interference Filters** | | | |
| manufacturer | Barr Associates | Barr Associates | Barr / Williams |
| peak transmission @ $\lambda_{on}/\lambda_{off}$ | 73/52 % | 50/65 % | 55/65 % |
| FWHM @ $\lambda_{on}/\lambda_{off}$ | 1.1/0.92 nm | 5/2 nm | 1/1 nm |
| **Photon Counting** | | | |
| photo-multipliers | Hamamatsu R7400P-03 | EMI 9893QA/350 | Hamamatsu R7400P-03, R9880U-110 |
| max count rate @ $\lambda_{on}/\lambda_{off}$ | 10/40 MHz | 6/2 MHz | 30/80 MHz |
| signal induced noise @ $\lambda_{on}/\lambda_{off}$ | 500 Hz/<20 Hz | <3 Hz/<0.3 Hz | <20 Hz/<10 Hz |
| range gating | all channels | none | not used [a] |
| mechanical chopper | 308 nm high gain | all | all |
| pre-amplifiers | - | none | 20x, 1.6 GHz |
| manufacturer | - | - | Becker&Hickl HFAC-26 |
| multi-channel scalers | Licel 300 MHz | Optech FDC 700 | FAST P7882-2, 200 MHz |

[a] implemented for all channels, but not used



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
