# Peer review of "Evaluation of the New DWD Ozone and Temperature Lidar during the Hohenpeißenberg Ozone Profiling Study (HOPS) and Comparison of Results with Previous NDACC Campaigns"

_Atmospheric Measurement Techniques, 2020_

## Referee Comment (RC1) · Anonymous Referee #1 · 29 Dec 2020

This paper is the second of two manuscripts describing recent intercomparisons of the NDAAC ozone and temperature lidars. The paper is generally sound and clear, but is fairly long with lots of figures and some repetition. Since the co-submitted manuscripts will be of interest to the same set of readers, more effort could be made to reduce the overlap between the two manuscripts. Section 6 (and Figures 14-17) of this paper could also be moved to a supplement to reduce the length of the main text.

Some specific comments.

The intercomparison described here has been formalized with a name: "The Hohenpeißenberg Ozone Profiling Study (HOPS)", but the name is not mentioned in either

the title or abstract and doesn't even appear in the text until page 3. It should appear in both.

P2, L35. What do LAVANDE and HOPE stand for? Spell it out somewhere.

P2, L36. The last half of this sentence is awkward.

P2, L45. "The previous NDACC campaign..." should be "A previous NDACC campaign..."

P3, L54. "During the recent NDACC validation campaign by Wing et al. ..." should read "During the more recent LAVANDE campaign (Wing et al., 2020)..."

P3, L59. "SABER" is defined in the abstract, but not the text.

P3, L66 and L79. Repetitive.

P3 ,L77. Replace "...35 km 2)..." with "...35 km, and 2)..."

P4,L82. This is the first appearance of "HOPS".

P4,L107. "...scattering cross-sections..." should read "...absorption cross-sections..."

P4,L108. Sentence could be shortened to "The first wavelength is generated using a 308 nm XeCl excimer laser."

P4, L114. "...ozone scattering targets..." should read "...ozone molecules..."

P4, L116. Awkward. Perhaps rephrase "Generating lidar temperature profiles is accomplished..." to "Lidar temperature profiles are derived..."

P4, L117. "...proton..." should read "...photon..."

P5, L123. Remove Table 4 callout, or re-order tables.

P5, L140. Repeats lines 122-123.

P6, L170. "...alos..." should be "...also..."

P6, L177. SABER should be defined before first usage (see comment for P3, L159).

P7, L190. "...principle..." should be "...principal..."

P7, L208. Superfluous comma.

P9, L225. Perhaps insert "(not shown)" after "both wavelengths"

P10, L241-2. Replace "An example of both..." with "Examples of ..." and insert ", respectively" after "...Figs. 3 and 4."

P12, L267+. I don't see the "tight clustering" at 40 km referred to in the text. The NASA and OMPS-uv measurements are consistently low at this altitude. At 30 km, the OMPS-vis measurements are consistently low so I don't think you can blame these differences on the temporal offset-particularly since the OMPS-uv measurements look OK. Any comment on the 21 Mar HOHO measurements at 20 km? This appears to be even larger outlier than the BM measurement at 30 km that the authors do single out. What happened to the NASA-STROZ measurements on this day?

P12, L286. Fig. 8 or Fig. 6?

P16, L294. Rephrase "The spread in of values..." as "The spread in the values..."

P16, L294+. The scatter plots in Figure 7 are hard to read and should be replotted with larger and different symbols (and axis labels). Perhaps use closed circles for the lidar data and open circles for the satellites? The statement about the MLS measurements being high at low values in the 15-20 km panel is not obvious from the plot. The most striking feature of this panel are the very low values measured by SABER, yet there is no mention of this in the text. As far as the scatter in the satellite measurements in the 20-30 km panel, why would fewer nights of observations necessarily lead to more scatter? Including the BM measurements in the 30-50 km panel seems inappropriate since the balloons rarely ascend past 35 km. Also, why would the wind displacements

cause these points to be consistently high-particularly since these displacements are still much smaller than most of the satellite paths? Isn't the BM pump a more likely culprit as you note below?

P18, L325. As before, where are the NASA-STROZ measurements for 21 Mar? Were there clouds that might have affected the HOH lidar measurements leading to the low O3 at 20 km and high T at 30 km?

P21, L360. What exactly is meant by "due to the combination of HOH lidar data with the radiosonde mentioned in Sect. 2.2."

P28, L453. "lidras" should be "lidars".

Figure 1 caption. DD-MM or MM-DD?

Figure 6. I assume the red trace labelled "HOH" is actually NASA-STROZ? In my opinion, ALL of the comparisons should be shown relative to the NASA-STROZ lidar and NOT the HOH lidar since it is the official NDAAC "traveling standard".

Figures 7 and 11. The different measurement sets are hard to discern in all of the scatter plots. The data should be replotted with different symbols and the axes labelled. I assume that the HOH measurements were used as the reference in Figures 6/7 and 11/12 because of the missing NASA data on 21 Mar. To me, the missing NASA measurements and outlying HOH ozone and temperature measurements on this day are something of a red flag, particularly since there is no explanation for either.

---

## Referee Comment (RC2) · Anonymous Referee #2 · 23 Mar 2021

General comments

This is a generally well written paper about an NDACC intercomparison campaign, augmented with results from other NDACC campaigns.

One of the main points in the paper is that the original Hohenpreissenberg lidar HOHO is compared to the NASA travelling reference lidar STROZ for the second time (2009 HOPE and 2018-2019 HOPS) and the new Hohenpreissenberg lidar HOH is now also compared to STROZ, as well as to the HOHO system. So the consistency of HOH with HOHO can be established on site for the overlapping parts of the profiles. It is
shown in Fig.2 that HOH has much greater performance in terms of range and SNR, but in the overlapping regions the systems are consistent. In order to validate HOH, STROZ is needed. Now, the general structure of the paper becomes sometimes a bit hard to follow, since the HOPS campaign are intertwined with LAVANDE results that have been published separately, which confronts the reader with a few storylines that have to be kept separate. The points brought in are certainly relevant, but it complicates the structure of the paper. In fact, this becomes clear in the conclusions: "The cross-comparison of NDACC campaign at Hohenpeißenberg Meteorological Observatory (HOPS) and at Observatoire de Haute Provence (LAVANDE) has allowed for the unique opportunity to assess potential biases in the NASA-STROZ reference lidar." My suggestion is to re-structure some of the sections to clarify this and move this material as much as possible to Sec.7.

Since the NDACC intercomparisons with a travelling reference lidar have been undertaken for some time (e.g. the references mentioned date back to 1995) it could be clearer described how the intercomparisons are generally carried out, according to an NDACC protocol, and perhaps explain how the HOPE and LAVANDE campaigns may be deviating from that protocol. There are some instances in the text that suggest there are different variants of the protocol. It would be interesting from the network design point of view to know why these variants exist.

Abstract - Remove the sentences "The previous 2017-2018 ... are reported in the companion article." - Add the main conclusions "The intercomparison exercise has confirmed that the original DWD lidar, HOHO continues to meet NDACC standards for ozone profiles at the 3% level between 16.5 and 43 km and at the 10% level between 10 and 44 km. The HOHO lidar meets the NDACC temperature standards for accuracy at the $\pm 1$ K level between 18 and 70 km. The new DWD lidar, HOH, meets the 3% ozone standard between 17 and 41 km, the 10% ozone standard between 15 and 41 km, and the $\pm 1$ K temperature standard 555 between 17 and 78 km." - Add "The cross-comparison of NDACC campaign at Hohenpeißenberg Meteorological Observatory (HOPS) and at Observatoire de Haute Provence (LAVANDE) has allowed for the unique opportunity to assess potential biases in the NASA-STROZ reference lidar. Possible biases may arise from algorithm initialisation choices and serve as strong motivation for another NDACC temperature algorithm paper."

Section 2 - The differences of the original Hohenpreissenberg lidar HOHO and the new lidar HOH are described. In the description of the travelling standard STROZ, it is not clear if instrument changes have been applied since the HOPE campaign in 2009. This is relevant since the consistency of the performance of HOHO is essentially compared again now in the HOPS campaign against the same travelling standard.

Section 6 - Earlier in the paper reference is made to Leblanc et al., 2016a, b, c. Are the results presented obtained using the methods described in those papers? Are results, following the blind intercomparison, processed by the proprietary processing algorithms of each group, or are they processed by a common processing code that is endorsed by NDACC? How would using a common code impact the intercomparison results for HOPE, HOPS and LAVANDE?

Small comments: - Not all readers may be familiar with Pearson's correlation coefficient. Pleaser briefly explain. - The names of the colours in the figures are a bit strange; e.g. "burnt orange", and "mustard". Why not just orange and yellow?

---

## Author Comment (AC1) · 30 Mar 2021

**Response to Reviewer 1 for AMT-2020-396**

Thank you very much for agreeing to review our article. I greatly appreciate your attention to detail and your challenges to our interpretation of the results. I hope that we have responded to all of your concerns in a satisfactory manner.

This paper is the second of two manuscripts describing recent intercomparisons of the NDAAC ozone and temperature lidars. The paper is generally sound and clear, but is fairly long with lots of figures and some repetition. Since the co-submitted manuscripts will be of interest to the same set of readers, more effort could be made to reduce the overlap between the two manuscripts. Section 6 (and Figures 14-17) of this paper could also be moved to a supplement to reduce the length of the main text.

**Specific comments:**
The intercomparison described here has been formalized with a name: "The Hohenpeißenberg Ozone Profiling Study (HOPS)", but the name is not mentioned in either the title or abstract and doesn't even appear in the text until page 3. It should appear in both.
Title is changed to:
"Evaluation of the New DWD Ozone and Temperature Lidar during the Hohenpeißenberg Ozone Profiling Study (HOPS) and Comparison of Results with Previous NDACC Campaigns"

L7 in the abstract replaced with:
"The campaign, referred to as the Hohenpeißenberg Ozone Profiling Study (HOPS), was conducted within the larger context of NDACC validation activities for European lidar stations."

P2, L35. What do LAVANDE and HOPE stand for? Spell it out somewhere.
Added (LidAr VAlidation NDacc Experiment) to L26
Added (Hohenpeißenberg Ozone Profiling Experiment) to L36

P2, L36. The last half of this sentence is awkward.
Sentence replaced with:
"When providing context for HOPS campaign we will refer back to the 2020 LAVANDE campaign (Wing et al., 2020) and the previous validation campaign at Hohenpeißenberg called HOPE (Hohenpeißenberg Ozone Profiling Experiment) (Steinbrecht et al., 2009)."

P2, L45. "The previous NDACC campaign. . ." should be "A previous NDACC campaign. . ."
Done.

P3, L54. "During the recent NDACC validation campaign by Wing et al. . . ." should read "During the more recent LAVANDE campaign (Wing et al., 2020). . ."
Done.

P3, L59. "SABER" is defined in the abstract, but not the text.
Added definition for both SABER and MLS on L60

P3, L66 and L79. Repetitive.
Removed from L66:
"A comparison was also conducted between NASA and the OHP temperature lidar LTA. The validation exercise determined that the photomultiplier in the low gain channel of LTA was defective and the component was subsequently replaced."

P3 ,L77. Replace ". . .35 km 2). . ." with ". . .35 km, and 2). . ."
Done.

P4,L82. This is the first appearance of "HOPS".
Fixed in response to general comments

P4,L107. ". . .scattering cross-sections. . ." should read ". . .absorption cross sections. . ."
Done.

P4,L108. Sentence could be shortened to "The first wavelength is generated using a 308 nm XeCl excimer laser."
Done.

P4, L114. ". . .ozone scattering targets. . ." should read ". . .ozone molecules..."
Done.

P4, L116. Awkward. Perhaps rephrase "Generating lidar temperature profiles is accomplished. . ." to "Lidar temperature profiles are derived. . ."
Done.

P4, L117. ". . .proton. . ." should read ". . .photon. . ."
Done.

P5, L123. Remove Table 4 callout, or re-order tables.
I'm using the AMT Latex template.  I will remember to ask the Copy editor about this.

P5, L140. Repeats lines 122-123.
L140 removed.

P6, L170. ". . .alos. . ." should be ". . .also. . ."
Done.

P6, L177. SABER should be defined before first usage (see comment for P3, L159).
Addressed in comment for P3, L159

P7, L190. ". . .principle. . ." should be ". . .principal. . ."
Done.

P7, L208. Superfluous comma.
Sentence split to read as follows:
For HOPS there are typically between 10 and 20 coincident profiles for each of the satellites. These profiles are generally divided between one or two satellite overpasses for a given night (the following morning for OMPS).

P9, L225. Perhaps insert "(not shown)" after "both wavelengths"
Done.

P10, L241-2. Replace "An example of both. . ." with "Examples of . . ." and insert ", respectively" after ". . .Figs. 3 and 4."
Done.

P12, L267+. I don't see the "tight clustering" at 40 km referred to in the text. The NASA and OMPS-uv measurements are consistently low at this altitude.
Text in L269 replaced with:
"The top panel which shows the ozone number densities at 40 km, indicates that in 2019 (last 8 nights) there was tight clustering of all the measurements except for OMPS which was consistently low and the NASA lidar which was significantly lower on three of the nights. During the 2018 portion of the campaign (first two nights) there was more variation between all instruments."

At 30 km, the OMPS-vis measurements are consistently low so I don't think you can blame these differences on the temporal offset-particularly since the OMPS-uv measurements look OK.
Good point.  Text replaced with:
"In the second panel, which shows ozone densities at 30 km, we see that there is tight clustering for all instruments except for the OMPS visible channel.  Given that the OMPS UV channel is in closer agreement with all of the other measurements and that the OMPS visible channel only extends to 35 km, it is probable that the observed low bias in OMPS visible is associated with the upper measurement limits of that channel."

Any comment on the 21 Mar HOHO measurements at 20 km? This appears to be even larger outlier than the BM measurement at 30 km that the authors do single out. What happened to the NASA-STROZ measurements on this day?
Added text:
"The NASA lidar experienced technical difficulties on the 21st of March 2019 and did not produce an ozone profile for the night.  Additionally, there was a substantial delay in starting the HOHO lidar compared to the HOH lidar. As a result, the HOHO nightly average profile was more heavily influenced by a transient ozone layer which was present on this night (not shown)."

P12, L286. Fig. 8 or Fig. 6? P16, L294. Rephrase "The spread in of values. . ." as "The spread in the values. . ."

Done.

P16, L294+. The scatter plots in Figure 7 are hard to read and should be replotted with larger and different symbols (and axis labels). Perhaps use closed circles for the lidar data and open circles for the satellites?

Figure 7 and 11 are replaced.

The statement about the MLS measurements being high at low values in the 15-20 km panel is not obvious from the plot.

Hopefully with the new plot markers it is easier to see that there are more dark purple circles below the 1:1 line than above it.

The most striking feature of this panel are the very low values measured by SABER, yet there is no mention of this in the text.

Added text:

"SABER (magenta) ozone number densities are significantly larger than all other measurements below 20 km and quickly become unreliable at lower altitudes (not shown)."

As far as the scatter in the satellite measurements in the 20-30 km panel, why would fewer nights of observations necessarily lead to more scatter?

Replaced with:

"The increased scatter may be due to geophysical variability."

Including the BM measurements in the 30-50 km panel seems inappropriate since the balloons rarely ascend past 35 km. Also, why would the wind displacements cause these points to be consistently high-particularly since these displacements are still much smaller than most of the satellite paths? Isn't the BM pump a more likely culprit as you note below?

Including the BM data above 30 km tells us something about how the instrument bias behaves at the limit of its measurement range.

Good point.  Changed to read:

"The right hand panel shows the scatter for all instruments from 30 to 50 km and we can clearly see the outlier data points in the Brewer-Mast (green) which likely arises from instrumental problems."

P18, L325. As before, where are the NASA-STROZ measurements for 21 Mar? Were there clouds that might have affected the HOH lidar measurements leading to the low O3 at 20 km and high T at 30 km?

Addressed in general comments.

P21, L360. What exactly is meant by "due to the combination of HOH lidar data with the radiosonde mentioned in Sect. 2.2."

Apologies this is a Typo - corrected to Sect. 2.1

The HOH lidar uses local radiosondes to correct low altitude temperatures.  The lidar is not truly independent at these altitudes.

P28, L453. "lidras" should be "lidars".

Done.

Figure 1 caption. DD-MM or MM-DD?

Corrected to DD-MM

Figure 6. I assume the red trace labelled "HOH" is actually NASA-STROZ?

Corrected.

In my opinion, ALL of the comparisons should be shown relative to the NASA-STROZ lidar and NOT the HOH lidar since it is the official NDAAC "traveling standard".

When we wrote the LAVANDE article I chose to use the French lidar as the reference for ozone as the NASA lidar experienced technical problems with the excimer laser during a portion of the campaign and as a result had fewer overall nights of ozone measurements.  In retrospect it was a mistake and I should have kept NASA as the reference for both temperature and ozone.

The authors debated the merits of correcting this mistake in HOPS (having NASA as the reference for both temperature and ozone) or keeping the same format as LAVANDE (local lidar for ozone and NASA for temperature).  We decided that for ease of the intercomparison between the two campaigns we would keep the LAVANDE format.  Going forward this is a lesson learned.

Figures 7 and 11. The different measurement sets are hard to discern in all of the scatter plots. The data should be replotted with different symbols and the axes labelled.

These two plots have been remade.

I assume that the HOH measurements were used as the reference in Figures 6/7 and 11/12 because of the missing NASA data on 21 Mar.

More about a poor reference decision made in LAVANDE.

To me, the missing NASA measurements and outlying HOH ozone and temperature measurements on this day are something of a red flag, particularly since there is no explanation for either.

I hope that the added text provides more context and lowers the red flag.

---

## Author Comment (AC2) · 30 Mar 2021

**Response to Reviewer 2 for AMT-2020-396**

Thank you very much for your comments on our paper. You have raised several important 'big picture' questions regarding NDACC intercomparison campaigns. I have tried to respond to several specific points. Many of your comments provide us with good motivations for the next paper.

This is a generally well written paper about an NDACC intercomparison campaign, augmented with results from other NDACC campaigns. One of the main points in the paper is that the original Hohenpreissenberg lidar HOHO is compared to the NASA travelling reference lidar STROZ for the second time (2009 HOPE and 2018-2019 HOPS) and the new Hohenpreissenberg lidar HOH is now also compared to STROZ, as well as to the HOHO system. So the consistency of HOH with HOHO can be established on site for the overlapping parts of the profiles. It is shown in Fig.2 that HOH has much greater performance in terms of range and SNR, but in the overlapping regions the systems are consistent. In order to validate HOH, STROZ is needed.

**General Comments:**
Now, the general structure of the paper becomes sometimes a bit hard to follow, since the HOPS campaign are intertwined with LAVANDE results that have been published separately, which confronts the reader with a few storylines that have to be kept separate. The points brought in are certainly relevant, but it complicates the structure of the paper. In fact, this becomes clear in the conclusions: "The cross-comparison of NDACC campaign at Hohenpeißenberg Meteorological Observatory (HOPS) and at Observatoire de Haute Provence (LAVANDE) has allowed for the unique opportunity to assess potential biases in the NASA-STROZ reference lidar."

My suggestion is to re-structure some of the sections to clarify this and move this material as much as possible to Sec.7. Since the NDACC intercomparisons with a travelling reference lidar have been undertaken for some time (e.g. the references mentioned date back to 1995) it could be clearer described how the intercomparisons are generally carried out, according to an NDACC protocol, and perhaps explain how the HOPE and LAVANDE campaigns may be deviating from that protocol.
We sympathize with the complexity for the reader - perhaps we have attempted too much for a single article. However, we are hesitant to make major structural changes to the paper as doing so would likely add text to an already long article.

There are some instances in the text that suggest there are different variants of the protocol. It would be interesting from the network design point of view to know why these variants exist.
The last review of NDACC lidar intercomparison practices was done 17 years ago (Keckhut et al., 2004). There is a great deal of work that *should* be done to develop and modernize NDACC "standard practices".

**Specific Comments:**

Abstract
- Remove the sentences "The previous 2017-2018 ... are reported in the companion article."
Done.

- Add the main conclusions "The intercomparison exercise has confirmed that the original DWD lidar, HOHO continues to meet NDACC standards for ozone profiles at the 3% level between 16.5 and 43 km and at the 10% level between 10 and 44 km. The HOHO lidar meets the NDACC temperature standards for accuracy at the ±1 K level between 18 and 70 km. The new DWD lidar, HOH, meets the 3% ozone standard between 17 and 41 km, the 10% ozone standard between 15 and 41 km, and the ±1 K temperature standard 555 between 17 and 78 km."
Added text to L13:
"Both the original and new DWD lidars continue to meet the NDACC standard for lidar ozone profiles by exceeding 3% accuracy between 16.5 and 43 km."

A similar statement for temperature is in the following paragraph.

- Add "The cross-comparison of NDACC campaign at Hohenpeißenberg Meteorological ObC2 servatory (HOPS) and at Observatoire de Haute Provence (LAVANDE) has allowed for the unique opportunity to assess potential biases in the NASA-STROZ reference lidar. Possible biases may arise from algorithm initialisation choices and serve as strong motivation for another NDACC temperature algorithm paper."
Last two paragraphs of the abstract are rewritten as:
"There was good agreement between all ozone lidar measurements in the range of 15 to 41 km with relative differences between co-located ozone profiles of less than $\pm$10\%.  Differences in the measured ozone numbers densities between the lidars and the locally launched ozone sondes were also generally less than 5\% below 30 km.  The satellite ozone profiles demonstrated some differences with respect to the ground based lidars which are due to sampling differences and geophysical variation. Both the original and new DWD lidars continue to meet the NDACC standard for lidar ozone profiles by exceeding 3\% accuracy between 16.5 and 43 km. Temperature differences for all instruments were less than $\pm 5 K$ below 60 km, with larger differences present in the lidar-satellite comparisons above this region. Temperature differences between the DWD lidars met the NDACC accuracy requirements of $\pm$1 K between 17 and 78 km.

A unique cross-comparison between the HOPS campaign and a similar, recent campaign at Observatoire de Haute Provence (LAVANDE) allowed for an investigation into potential biases in the NASA-STROZ reference lidar. The reference lidar may slightly underestimate ozone number densities above 43 km with respect to the French and German NDACC lidars.  Below 20 km the reference lidar temperatures profiles are 5 to 10 K cooler than the temperatures which are reported by the other instruments.  The differences in both temperature and ozone are likely due to choices in data treatment."

Section 2

- The differences of the original Hohenpreissenberg lidar HOHO and the new lidar HOH are described. In the description of the travelling standard STROZ, it is not clear if instrument changes have been applied since the HOPE campaign in 2009. This is relevant since the consistency of the performance of HOHO is essentially compared again now in the HOPS campaign against the same travelling standard.

Components of the NASA lidar have been replaced over time but the experimental design of the travelling standard lidar is the same.  It is impossible to guarantee that the standard remains absolutely unchanged over 25 years as there is no external reference.  It is encouraging that in both the 2008 HOPE campaign and the 2019 HOPS campaign NASA measured slightly less ozone above approx. 45 km than the DWD lidar.

Section 6
- Earlier in the paper reference is made to Leblanc et al., 2016a, b, c. Are the results presented obtained using the methods described in those papers? Are results, following the blind intercomparison, processed by the proprietary processing algorithms of each group, or are they processed by a common processing code that is endorsed by NDACC? How would using a common code impact the intercomparison results for HOPE, HOPS and LAVANDE?

 The three Leblanc papers provide a common set of definitions and metrics for NDACC ozone and temperature profiles.  However, each group uses their own codes and analysis procedures on their lidar measurements.  The blind intercomparison only identifies the absolute differences between the final data products.

A common NDACC processing code for use during intercomparison campaigns is a very good idea.  It would remove a source of uncertainty from the interpretation of our results.  Additionally, comparing the measurements from each lidar as processed by both codes would allow for the separation of instrumental and computational sources of bias.

We hope that this article offers some motivation for the NDACC community to agree to the development of such a "standard intercomparison code".

Small comments:
- Not all readers may be familiar with Pearson's correlation coefficient. Please briefly explain.
Added to L370:
"(a measure of linear correlation between two datasets)"

- The names of the colours in the figures are a bit strange; e.g. "burnt orange", and "mustard". Why not just orange and yellow?
I used the names for the standard colours in the Matlab RBG colour palette. "Burnt orange" and "mustard" have more brown in them (look darker) than orange or yellow.